



# Detecting the human fingerprint in the summer 2022 West-Central European soil drought

Dominik L. Schumacher[1], Mariam Zachariah[2], Friederike Otto[2], Clair Barnes[2], Sjoukje Philip[3], Sarah Kew[3,] Maja Vahlberg[4], Roop Singh[4], Dorothy Heinrich[4], Julie Arrighi[4,5,6], Maarten van Aalst[4,6,7], Mathias Hauser[1], Martin Hirschi[1], Verena Bessenbacher[1], Lukas Gudmundsson[1], Hiroko K. Beaudoing[8,9], Matthew Rodell[8], Sihan Li[10], Wenchang Yang[11], Gabriel A. Vecchi[11,12], Luke J. Harrington[13], Flavio Lehner[14,15], Gianpaolo Balsamo[16], and Sonia I. Seneviratne[1]

[1] Institute for Atmospheric and Climate Science, ETH Zurich, 8092, Switzerland

[2] Grantham Institute, Imperial College, London, SW7 2BU, UK

[3] Royal Netherlands Meteorological Institute (KNMI), De Bilt, 3731, The Netherlands

[4] Red Cross Red Crescent Climate Centre, The Hague, 2593, Netherlands

[5] Global Disaster Preparedness Center, Washington DC, 20006, USA

[6] University of Twente, Enschede, 7500, Netherlands

[7] International Research Institute for Climate and Society, Columbia University, New York, NY 10964-1000, US

[8] Earth Sciences Division, NASA GSFC, Greenbelt, MD 20771, USA

[9] Earth System Science Interdisciplinary Center, University of Maryland, College Park, MD 20740, USA

[10] Department of Geography, University of Sheffield, S10 2TN, UK

[11] Department of Geosciences, Princeton University, Princeton, NJ 08544, USA

[12] High Meadows Environmental Institute, Princeton University, Princeton, NJ 08544, USA

[13] Te Aka Mātuatua School of Science, University of Waikato, Hillcrest, Hamilton 3214, New Zealand

[14] Department of Earth and Atmospheric Sciences, Cornell University, Ithaca, NY 14853-1504, USA,

[15] Climate and Global Dynamics Laboratory, National Center for Atmospheric Research, Boulder, CO 80301, USA

[16] European Centre for Medium-range Weather Forecasts, ECMWF, Reading, RG2 9AX, UK

**Corresponding author**: Dominik L. Schumacher (dominik.schumacher@env.ethz.ch)

**Abstract.** In the 2022 summer, West-Central Europe and several other northern-hemisphere mid-latitude regions experienced substantial soil moisture deficits in the wake of precipitation shortages and elevated temperatures. Much of Europe has not witnessed a more severe soil drought since at least the mid-20th century, raising the question whether this is a manifestation of our warming climate. Here, we employ a well-established statistical approach to attribute the low 2022 summer soil moisture to human-induced climate change, using observation-driven soil moisture estimates and climate models. We find that in West-Central Europe, a June–August root-zone soil moisture drought such as in 2022 is expected to occur once in 20 years in the present climate, but would have occurred only about once per century during pre-industrial times. The entire northern extratropics show an even stronger global warming imprint with a 20-fold soil drought probability increase or higher, but we note that the underlying uncertainty is large. Reasons are manifold, but include the lack of direct soil moisture observations at the required spatiotemporal scales, the limitations of remotely sensed estimates, and the



resulting need to simulate soil moisture with land surface models driven by meteorological data. Nevertheless, observation-based products indicate long-term declining summer soil moisture for both regions, and this tendency is likely fueled by regional warming, while no clear trends emerge for precipitation. Finally, our climate model analysis suggests that in a 2 °C world, 2022-like soil drought conditions would become twice as likely for West-Central Europe compared to today, and

would take place nearly every year across the northern extratropics.

## 1 Introduction

Following a dry spring with above-average air temperatures across much of Europe (Toreti et al., 2022), the 2022 summer was assessed "hottest on record" by the European Union's Copernicus environmental programme. The unusually hot and dry conditions were accompanied by widespread soil desiccation, particularly in western regions of the continent (Copernicus,

2022a) that experienced a sequence of heatwaves (Zachariah et al., 2022) and precipitation shortages. Based on runoff anomalies, it was highlighted in the press that the 2022 European drought could be the "worst in 500 years" (Henley, 2022). This event was preceded by the 2018–2020 drought in Europe (e.g., Boergens et al., 2020, Rakovec et al., 2022), and while 2021 brought relief to dry soils through above-normal precipitation in western parts of the continent (Copernicus, 2021), soil moisture drought indicators point to an incomplete recovery in many areas (NASA GRACE-FO, 2022; EDO, 2022). As

such, at least part of Europe was already primed for a severe soil drought in 2022 well before summertime. Unusual heat and drought also characterised the 2022 boreal summer elsewhere, however; for example, China was affected by exceptionally high aridity and temperatures (Ahmedzade et al., 2022), and North America experienced a warm summer with below-average soil moisture (Copernicus, 2022a). In the midlatitudes, extreme summer heat and precipitation shortages are typically fostered by persistent, often near-stationary anticyclones (e.g., Li et al., 2020), or in some cases subtropical ridges

(e.g., Sousa et al., 2020), and many areas in Europe were indeed subject to the strongest 500 hPa geopotential height anomalies between May and July 2022 since 1950 (Toreti et al., 2022). Such anticyclonic circulation patterns are intrinsically related to the extratropical jetstream, which is known to simultaneously promote drought and heavy precipitation in different regions for certain (wavy) flow configurations (Lau & Kim, 2012, Coumou et al., 2014), but the underlying dynamics are complex and still not fully understood. Nevertheless, a recent study has suggested that many

heatwaves in the ongoing century in Western Europe have been caused by increasingly frequent and persistent double jets, whose occurrence is closely linked to anticyclonic flow (Rousi et al., 2022). From a global perspective, ENSO has remained in the "La Niña" phase since late 2020 (CPC, 2022), which may have contributed to the hot and dry conditions in parts of both China and North America (Wang et al., 2007, Karori et al., 2013).

While the roles of such and other local and remote dynamic and thermodynamic drivers for the dry and hot 2022 summer are

yet to be investigated in detail, it is already clear that the soils in large parts of the northern extratropics were unusually dry. As such, enhanced land–atmosphere coupling (e.g., Seneviratne et al., 2006, Mueller and Seneviratne 2012, Miralles et al., 2019, Stegehuis et al., 2021) likely contributed to heatwaves in Europe, China, the southwestern United States (NASA Earth Observatory, 2022), and triggered hot and dry summer conditions in large parts of the northern extratropics. On the other





hand, the high temperatures likely exacerbated dry soil conditions due to increased land evapotranspiration, as identified in
recent drought events in Europe (e.g. Seneviratne et al. 2012, Teuling et al. 2013). This is in line with a detected trend
towards decreased water-availability in the dry season across land regions in the recent past, 1985–2014, compared to the
first half of the 20th century (Padrón et al., 2020). Furthermore, the mechanism of northward 'drought propagation' — a
causal link between (spring) drought in the Mediterranean, and hot and dry summers in West-Central Europe (Vautard et al.,
2005; Zampieri et al., 2009) — may also have played a role in the evolution of the 2022 European drought. The extreme
conditions manifested in some of the most severe soil moisture droughts on record; e.g., in July 2022, nearly half of Europe
was assigned a drought warning (EC JRC, 2022a). In some areas, shortages of drinkable water due to low water tables were
reported, whereas China issued its first nationwide drought alert (Reuters, 2022). In addition, the combination of excessive
heat and moisture deficits strongly increased the fire risk in Europe, leading to the highest burnt area ever recorded since the
start of measurements (EFFIS, 2022).

Low soil moisture typically implies increased water stress for natural vegetation and crops (e.g., Berg & Sheffield, 2018, Liu
et al., 2020), which can be further exacerbated by elevated air temperatures and hence heat stress (Seneviratne et al., 2021).
According to the 6th Assessment Report from the Intergovernmental Panel on Climate Change (IPCC), there is *medium
confidence* that human-induced climate change has contributed to increases in agricultural and ecological droughts in some
regions due to evapotranspiration increases (Seneviratne et al., 2021). Nevertheless, while strong evidence for human-
induced aggravations of recent heatwaves has been reported repeatedly (Seneviratne et al., 2021), such as for the heatwave in
Western Europe in July 2022 (Zachariah et al., 2022), there are more uncertainties in the contribution of anthropogenic
climate change to trends in agricultural drought conditions in single regions. Building on a rapid attribution analysis of the
World Weather Attribution (Schumacher et al., 2022), we investigate the role of climate change in the frequency and
magnitude of 2022 surface and root-zone soil moisture deficits — the latter a measure of agro-ecological drought — for two
regions: the West-Central Europe (WCE) region in IPCC AR6 (Iturbide et al., 2020), and the northern extratropics, i.e. the
land area between 23.5 °N to 90 °N (NHET). We restrict our analysis to boreal summer (June–August), the season with the
largest spatial extent of droughts in the northern extratropics (Lu et al., 2019), which is also when the widespread 2022
drought conditions peaked. As temperature and precipitation anomalies are known to strongly influence agricultural drought,
we also analyse summertime mean temperature and precipitation over the same regions as for soil moisture.

**2 Event description and associated impacts**

Several regions across the northern extratropics suffered from persistent drought and heatwaves in the 2022 summer. Parts of
southwestern North America, for example, were reported to experience their driest period in more than 1,200 years, causing
three water reservoirs in northern Mexico to drain and leading to water insecurity for five million residents (Linthicum,
2022). China, and particularly Hunan Province, experienced its longest drought and most severe heatwave on record (CMA,
2022; Ahmedzade et al., 2022; Le Page, 2022). As of 10 August 2022, nearly two-thirds of Europe was affected by drought





(Seabrook, 2022). We focus on West-Central Europe in the following and explore associated impacts in the context of vulnerability and exposure, since such extreme dry and hot conditions are known to act as a risk multiplier for energy, environmental and socio-economic vulnerability (Rakovec et al., 2022; Gazol and Camarero, 2022; Naumann et al., 2021).

Since the beginning of May to mid-September, five back-to-back heatwaves blanketed large swathes of Europe. Throughout
these months, several daily and monthly maximum temperature records were broken across Italy, France, Switzerland, Germany, Poland, Hungary, and Slovenia (see, e.g., Phys.org, 2022; Breteau, 2022; le News, 2022; Wang, 2022; Twoja Pogoda, 2022; OMSZ, 2022; BBC Weather, 2022). It is estimated that the persistent heat has led to over 24'000 fatalities across Europe, more than 18'000 of which within Western Central Europe — 11'000 in France and over 8'000 in Germany alone (Roucaute, 2022; Destatis, 2022a; Destatis, 2022b). Infrastructure was also impacted as the heat melted roads, buckled
railway lines, halted public transportation services, and increased the electricity demand while power stations operated at reduced capacity (Dhanesha & Jones, 2022; Binnie & Twidale, 2022; Rocha, 2022). The hot and dry conditions were also associated with a spike in wildfires; by 24 September, more than 770.000 hectares of land had burnt throughout the European Union (EU; EFFIS, 2022), which equals nearly three times the EU average over 2006-2021 (Copernicus, 2022b). Italy, Slovenia, France, and Romania were particularly affected by these fires ( Roscoe, 2022; Lukov, 2022; Korosec, 2022;
Dumitrescu, 2022), and by late June, Italy had crushed its historical wildfire average threefold (The Local, 2022).

Europe's prolonged hot and dry weather conditions during the first half of 2022, and ensuing low water reservoir levels, has led to significant reductions in summer crop yields, most significantly in France, northern Italy, Germany, Slovenia, Hungary, and Romania (EC JRC, 2022b). These significant agricultural impacts are unsurprising, given that this sector is the most water intensive industry in the region (Heggie, 2020; EEA, 2020a). Compared to their five-year averages, maize,
soybean and sunflower crops suffered 16, 15 and 12 percent decreases, respectively (Toreti et al., 2022). For example, in northern Italy, the Po River basin experienced its worst water crisis in approximately 70 years, leading to an estimated 30 percent reduction of rice crop yields and at least 50 cattle deaths (Clifford, 2022; Coldiretti, 2022). Paired with the Ukraine crisis hiking up the price for fertilisers fourfold, these decreases in agricultural production lead to a "heatflation" of food prices as well as higher feed prices for livestock (DW, 2022; Mendes, 2022). Food prices in China, especially for fruits and
vegetables, were also driven up by the heat and drought (Bradsher & Dong, 2022). We note that crop loss poses increasing threats to food security not only in the affected regions, but also globally, and is hence one of the major impacts of the 2022 drought.

The drought in Europe also had indirect impacts on electricity generation in several European countries (Horowitz, 2022). Lower river flows and thus lower reservoir levels have significantly decreased hydroelectric power generation; for example,
in Italy, hydroelectric plants generated 40-50 percent less power over the summer months, and one plant in Piacenza was temporarily shut down (Good et al., 2022). The low water levels in rivers in Germany also reduced the ability to transport coal by boat, further impacting energy supply (Horowitz, 2022). In France, where nuclear energy provides a clear majority of



the electricity, decreased water availability and the associated lack of cooling mandated output reductions and complete
shutdowns of nuclear reactors on the Belgium-France border (Kollewe, 2022). These supply constraints coincided with high
demand, particularly due to air conditioning during hot periods. We also emphasise here that the drought occurred at a time
when Europe was facing a number of other, compounding stressors on its energy supply; the COVID-19 pandemic led to a
slow-down in demand for energy in 2020, but demand has rebounded by 2022 while supply has not kept up, leading to an
increase in global energy prices. In addition, the war in Ukraine has strained ties between Europe and Russia, until recently
the main supplier of Europe's natural gas. The restricted supply sent prices soaring, with different impacts across European
countries based on their energy mix and import capacity from alternative routes. In Germany, for instance, the energy crisis
had (and still has) far-reaching economic ramifications affecting small and medium-sized enterprises, the backbone of
Germany's economy (Kagerl et al., 2022).

The drought also highlighted the vulnerability of Europe's water infrastructure; roughly 66% of the European population
relies on groundwater for its water-related needs, about 60% are residing in cities where groundwater is over-exploited
(EEA, 2020b), and water wastage in public supply systems is estimated at 20–40% of the available water for the entire EU
and up to 80% in individual cities (EEA 2020b; Hirschnitz-Garbers et al., 2016). In response to the 2022 drought, and to
effectively conserve water, multiple countries enforced water protection practices; e.g., cities and districts across Germany
prohibited extraction from various bodies of water, as well as filling pools, watering lawns, and cleaning cars (Stresing and
Wolf, 2022). By early August 2022, over 100 French municipalities relied on water deliveries by truck to overcome potable
tap water shortages (Chadwick, 2022), and 62 of 96 of France's departments were at the highest level of drought alert, many
of which implemented water restrictions (Al Jazeera, 2022). While the country experienced its driest July in more than half a
decade (BBC, 2022), and its reportedly most extreme drought in history (Breeden, 2022), the effects of the below-average
rainfall was likely aggravated by unsustainable water use and losses in the water distribution system.

We also point out that until recently, drought risk management at the pan-European scale has predominantly focused on
coping with financial losses, mainly through calamity funds, mutual funds, and insurances (Bielza Diaz-Caneja et al., 2009).
This opposes the current scientific consensus, which entails a shift from reactive to proactive risk management strategies
(Wilhite et al., 2007; Blauhut et al., 2016). Blauhut et al. (2022) note that drought risk management planning does not exist
on a unified continental scale in Europe, despite the potential benefits for large-scale directive planning in reducing
emergency response costs. Following a comprehensive review of drought management practices in 28 European countries
and surveying 712 experts across Europe, the paper recommends some key areas to reduce vulnerability and exposure from
the planning perspective. Its recommendations include developing a pan-European approach to drought management,
allowing for country contextualization while also supporting cross-border drought preparedness efforts.

To summarise, the 2022 heat and drought in Western Central Europe had far-reaching impacts on a variety of sectors
including health, energy, agriculture, and municipal water supply, reflecting the need to reassess drought preparedness and



deal with trade-offs in water management. It came at a time when its impacts were interacting with non-climate risks to create compounding and cascading impacts. For example, impacts on power generation due to heat and drought (on hydropower, nuclear and coal power plants) coincided with increasing energy prices linked to the conflict in Ukraine. Similarly, impacts on agricultural yields in Europe coincided with strained global food supply due to reduced exports from Russia and Ukraine, as well as high fertiliser prices with knock-on effects on inflation in Europe, but also on global food

prices and therefore food insecurity, resulting in risks cascading across sectors and regions (as flagged as a rising risk in IPCC AR6 WGII; IPCC 2022). Overall, this event serves as a strong motivation to strive for proactive drought risk management strategies, and developing and coordinating such strategies across the continent is considered an effective approach to improve drought preparedness and resilience.

## 3 Data and methods

We examine trends in root zone soil moisture over the two selected regions — WCE and NHET — for quantifying the role of climate change in the widespread 2022 drought conditions that impacted large parts of the northern extratropics, and in particular the European continent. We also compare the results for these variables and regions with the respective estimates based on precipitation and temperature to gain insights on how the soil moisture drought has been influenced by the accompanying precipitation deficits and anomalously high temperatures.

### 3.1 Observational data

### 3.1.1 Main datasets

To analyse the drought event, we rely on a mixture of reanalysis and observation-based data. The employed statistical approach (Sect. 3.3), mandates (i) continuous data, ideally since pre-industrial times but at least from 1950 onwards (van Oldenborgh et al., 2021), and (ii) data coverage at least until August 2022 to infer the probability — or return period — of

the 2022 summer soil drought. Many available soil moisture observations do not meet these criteria, hence we perform our main analysis for 1950–2022 and with soil moisture estimates derived from land surface models ingesting reanalysed or observed meteorological data, a common approach for assessing soil moisture trends (e.g., Albergel et al., 2013; Bi et al., 2016; Cheng & Huang, 2016; Deng et al., 2020; Qiao et al., 2021; Almendra-Martín et al., 2022). These datasets are introduced below along with additional meteorological variables used for analysis.

ERA5

The ERA5 reanalysis product by the European Centre for Medium-Range Weather Forecasts (ECMWF) contains simulated estimates of climate variables for the period 1950-present, at 0.25 km × 0.25 km resolution and at hourly intervals (Hersbach et al., 2020). ERA5 uses the ECMWF assimilation system IFS (IFS Cycle 41R2), and simulates land surface processes with the Hydrology Tiled ECMWF Scheme for Surface Exchanges over Land (H-TESSEL) model. For the production of ERA5,

more than 200 satellite instruments and conventional meteorological data types are assimilated, including scatterometer soil



moisture, rain gauge–radar composites, and in-situ measurements of 2m temperature and humidity. We use the data of four variables: precipitation, temperature and volumetric soil moisture at surface (0cm–7cm) and root zone levels (0cm–100cm). ERA5 soil moisture is computed for four vertical soil layers extending down to 289cm, of which the first three are aggregated here to the root-zone soil moisture. Due to unrealistic values in Greenland, especially noticeable prior to ~1970,

we mask the affected area prior to calculating the regional mean of the northern extratropics. For other datasets without problematic soil moisture values in Greenland, including or masking the latter results in nearly identical timeseries as Greenland (about 2*10^6 km2) only accounts for a small fraction of the entire northern extratropical land area (on the order of 80*10^6 km2).

Together with the other main soil moisture datasets (and additionally EFAS-historical for West-Central Europe), we use

ERA5 data for our comparison of summertime root-zone soil moisture. However, we refrain from employing ERA5 root-zone soil moisture for the statistical analysis because ERA5 has been produced using several production streams (see Table 3 in Hersbach et al., 2020), employing a spin-up period of 1 year for merging the different simulations. This is known to cause discontinuities in the deep soil, and manifests in visible jumps in NHET root-zone soil moisture (not shown). Even though the effects on WCE root-zone soil moisture are not as obvious, we exclude ERA5 root-zone soil moisture for both of our

domains and rely on ERA5-Land instead. Surface soil moisture is not affected, as its memory timescale is on the order of days to weeks (McColl et al., 2019), and is thus included in the additional surface soil moisture attribution analysis.

### ERA5-Land

ERA5-Land is an offline 0.1° × 0.1° land surface model simulation that ingests ERA5 precipitation and altitude-corrected air temperature, humidity and pressure to match the higher resolution land grid (Muñoz-Sabater et al., 2021). Even though the

employed land surface model H-TESSEL represents land processes similarly as within IFS, the non-linear downscaling enables a more realistic simulation of the hydrological cycle in ERA5-Land compared to ERA5. We also note that ERA5-Land has been produced with only 3 production streams, and that these streams are merged with a spin-up period of 3 years (rather than 1 year as for ERA5). While minor discontinuities are still evident in the deepest layer of ERA5-Land at 100–289 cm (not shown), the root-zone as defined in this study (0-100 cm) is not affected. As such, ERA5-Land does not feature the

same collation issues as ERA5 (see above), and is thus more suitable for our attribution study. In addition, restricting the analysis to ERA5-Land and GLDAS-CLSM prevents an overrepresentation of ECMWF products, since ERA5 and ERA5-Land are (by design) closely related.

### GLDAS-CLSM

We also employ the NASA Global Land Data Assimilation System Catchment Land Surface Model (GLDAS-CLSM; Rodell

et al., 2004; Li et al., 2019). Initialised using the soil moisture and spatial fields from the land surface model climatology for January 1, 1948, the simulations are forced by the global meteorological forcing data from Princeton University (Sheffield et al., 2006) and (after 2003) by ECMWF IFS analysis fields. Run within the Land Information System (LIS; Kumar et al., 2016) framework, this model simulates water storage in the full soil profile at 0.25° × 0.25° resolution, from which surface



(0-2 cm) and root zone (0-100 cm) soil moisture and groundwater can be derived. Outputs are available from 1948 to the
present. Beginning in 2003, the model assimilates GRACE/GRACE-FO terrestrial water storage anomaly data from the
University of Texas (Save et al., 2016; Save, 2020)). The 2003 to present climatology is scaled to the open loop, Princeton-
forced (1948-2012) climatology prior to computing drought/wetness indicators (Houborg et al., 2012). For some grid cells in
high latitudes, this results in negative and hence not physically meaningful values, which we remove for our analysis of the
northern extratropics. Note that the GLDAS-CLSM dataset employed here was produced with GRACE/GRACE-FO data up
to and including June 2022. We do not expect this to affect our conclusions, since (i) the attribution statements ultimately
depend on the linear trend in soil moisture as a function of global warming for the entire period of 1950-2022, and (ii) the
return periods of the 2022 event are similar in ERA5-Land and GLDAS-CLSM.

E-OBS

For the WCE region, we additionally analyse E-OBS (version 25.0e). The E-OBS dataset is a 0.25° × 0.25° gridded
temperature and precipitation dataset of Europe, formed from the interpolation of station derived meteorological
observations (Cornes et al., 2018). E-OBS was used to produce seasonal cycles and climatology and for trend analysis of
precipitation and temperature over Europe.

GISTEMP

Finally, as a measure of anthropogenic climate change, we use the global mean surface temperature (GMST from the
National Aeronautics and Space Administration (NASA) Goddard Institute for Space Science (GISS) surface temperature
analysis (GISTEMP, Hansen et al., 2010 and Lenssen et al. 2019). GMST is low-pass filtered with a 4-year running mean.

### 3.1.2 Supplementary datasets

We emphasise that the soil moisture in reanalyses and land surface model simulations is a derived variable affected both by
model formulation and the quality of meteorological forcing (and, in the case of ERA5, the assimilated soil moisture data).
Compared to meteorological variables such as temperature or precipitation with far better in-situ coverage, soil moisture
estimates are hence associated with considerable uncertainty. In this context, it is worth noting that in recent decades and
particularly in the ongoing millennium, the progressive deployment of satellites and development of more capable sensors
has ushered in an era of remote-sensed surface soil moisture estimates. However, microwave remote-sensed soil moisture
products typically feature data gaps due to incomplete satellite coverage and radio frequency interference as well as
environmental conditions that prevent the measurement of soil water content such as dense canopies, frozen soil or snow
cover (Llamas et al., 2020, Bessenbacher et al., 2022, Liu et al., 2023). Nevertheless, satellite-based soil moisture estimates
can — if adequately gap-filled — provide a valuable alternative perspective for assessing recent large-scale surface soil
moisture changes (Bessenbacher et al., in prep.). The advances in remote sensing are complemented by the development of
increasingly capable approaches in the field of artificial intelligence to extract the most out of the available data, enabling
additional lines of evidence. We thus expand our analysis by using data obtained with various observation-driven approaches



listed below, including soil moisture estimated by comparatively simple process-based models or neural networks instead of land surface models, ingesting both in-situ and remote-sensed measurements.

### EFAS-historical

The European Drought Observatory provides information on the current status of drought in Europe, including a soil
moisture index (SMI) and SMI anomalies based on the European Flood Awareness System (EFAS). The latter is a hydrological forecasting and monitoring system from the European Commission and the ECMWF, ingesting a range of meteorological forecasts at medium to seasonal timescales as well as observations. The underlying hydrological model is LISFLOOD (EC JRC, 2020), a hydrological rainfall-runoff model, and we employ EFAS-historical simulations (Mazetti et al., 2020) forced with meteorological observations and available every 6 hours at 5 km × 5 km for Europe since 1991. Soil
moisture is provided at three soil levels (superficial, upper and lower soil), yet these depths vary for each grid cell and are not provided. While soil evaporation is restricted to the superficial layer, plant roots can extract moisture from both the superficial and upper soil layer for transpiration. As the lack of layer depth information prevents vertical aggregation, we rely on the upper soil as a proxy for root-zone soil moisture in our study, whereas the superficial layer represents the surface soil moisture.

### SoMo.ml

Generated with a Long Short-Term Memory neural network ingesting in-situ measurements and ERA5 meteorological forcing, SoMo.ml provides global daily soil moisture data from 2000 to 2019 at 0.25° × 0.25° horizontal resolution (O. & Orth, 2021). Since the in-situ soil moisture data collected across more than 1000 sites is based on several sensor types and different calibrations, the creators of SoMo.ml employ ERA5 soil moisture to scale the point-scale measurements. In
essence, the mean and standard deviation of soil moisture is inferred from ERA5, whereas point-level in-situ data represent the temporal dynamics. With this approach, the machine learning model can be trained around the globe to estimate soil moisture at the grid-scale rather than only for individual sites. The resulting data are provided at three depths (0 cm–10 cm, 10 cm–30 cm and 30 cm–50 cm), and we use the uppermost layer as an indicator of surface soil moisture. We note that the performance of this dataset depends on in-situ data availability, and is hence limited in sparsely monitored areas such as the
tropics.

### RSSSM

The remote-sensing-based surface soil moisture (RSSSM, Chen et al., 2020) combines 11 high-quality microwave products with a neural network approach, resulting in gap-free, global data. This dataset has been validated against in-situ measurements, and is provided at 0.1° × 0.1° every 10 days from 2003 to 2018.

### ESA-CCI gap-filled

The European Space Agency (ESA) Climate Change Initiative (CCI) soil moisture products are currently (v07.1) based on either 5 active, 12 passive or a blend of all (microwave) sensors, and provide remote-sensed surface soil moisture estimates



since 1978 (Gruber et al., 2019; Preimesberger et al., 2021). In these products and due to the difficulties outlined above, however, less than half of all global land data points are observed in the years 2003-2020 (Bessenbacher et al., in prep.). In

addition, trend analyses are complicated by the fact that the sensor coverage is not constant in time and used to be fairly limited; e.g., a majority of the northern extratropics is only covered since 2007 (Dorigo et al., 2017). To address these problems, a gap-filled soil moisture product is currently being developed by ESA-CCI, building on the combined product (blended active + passive) and the application of the DCT-PLS smoothing algorithm (Garcia, 2010) for the gap filling. In addition, GLDAS-Noah v2.1 surface soil temperature data is used for the detection of frozen soil conditions, in which case

soil moisture values are gap-filled by a temporal linear interpolation. The gap-filled daily product at 0.25° x 0.25° is currently available for 2000–2021.

ESA-CCI gap-filled with the multivariate CLIMFILL approach

The recently developed CLIMFILL is a multivariate gap-filling framework (Bessenbacher et al., 2022) that exploits the spatial, temporal, and cross-variable dependence structure of Earth system observations. It has been used to gap-fill a wide

range of observations including surface temperature, precipitation and ESA CCI surface soil moisture, on monthly grids from 1995 to 2020 and at 0.5° × 0.5° spatial resolution (Bessenbacher et al., in prep.). Within this dataset, gaps in surface soil moisture are filled by taking into account information acquired through spatial interpolation of the monthly maps, temporal lagged effects like soil moisture memory and observed values of related variables at the land surface, for example temperature and precipitation. Bessenbacher et al. (in prep.) have demonstrated that this approach fills gaps in the data more

accurately than univariate interpolation that cannot take into account information from other observed variables.

**3.2 Model and experiment descriptions**

We use several climate modelling experiments in this study, consisting of three multi-model ensembles using different framings (Philip et al., 2020a): coupled global circulation models (GCMs), sea surface temperature (SST) driven GCMs, and high resolution models.

The first set of models used in this analysis comes from the CMIP6 experiment (Eyring et al., 2016). For all simulations, the period 1850 to 2015 is based on historical coupled simulations, while the SSP5-8.5 scenario is used for the remainder of the 21st century. Models are excluded if they do not provide all relevant variables, do not cover 1850–2100, or include duplicate time steps or missing time steps. The first available ensemble member is used for each model.

The second set of models used in the analysis include the AM2.5C360 (Yang et al., 2021; Chan et al. 2021) and the FLOR

(Vecchi et al., 2014) high-resolution climate models developed at Geophysical Fluid Dynamics Laboratory (GFDL). The AM2.5C360 is an atmospheric GCM based on that in the FLOR model (Delworth et al., 2012, Vecchi et al., 2014) with a horizontal resolution of 25 km. Ten ensemble simulations of the Atmospheric Model Intercomparison Project (AMIP) experiment (1871–2021) are analysed. These simulations are initialised from ten different pre-industrial conditions but





forced by the same SSTs from HadISST1 (Rayner et al., 2003) after groupwise adjustments (Chan et al., 2021), as well as
the same historical radiative forcings. The FLOR model, on the other hand, is an atmosphere-ocean coupled GCM with a
resolution of 50 km for land and atmosphere and 1° for ocean and ice. Ten ensemble simulations from FLOR are analysed,
which cover the period from 1860 to 2100 and include both the historical and RCP4.5 experiments driven by transient
radiative forcings from CMIP5 (Taylor et al., 2012).

The third ensemble considered in this study is the HighResMIP SST-forced model ensemble (Haarsma et al., 2016), the
simulations for which span from 1950 to 2050. The SST and sea ice forcings for the period 1950-2014 are obtained from the
0.25° x 0.25° Hadley Centre Global Sea Ice and Sea Surface Temperature dataset that have undergone area-weighted
regridding to match the climate model resolution. For the 'future' time period (2015-2050), SST/sea-ice data are derived
from RCP8.5 (CMIP5) data, and combined with greenhouse gas forcings from SSP5-8.5 (CMIP6) simulations (see Sect. 3.3
of Haarsma et al., 2016 for further details). It is worth noting that this ensemble only has outputs for moisture in the upper
portion of the soil column (i.e., the upper 10cm of the soil layer), but not moisture in the total soil column, therefore is not
considered in the analysis of root zone soil moisture.

### 3.3 Statistical methods

In this study we analyse summer (June–August) mean time series of soil moisture, precipitation and temperature, averaged
over both West-Central Europe and the northern extratropics, as defined in Sect. 1. Methods for observational and model
analysis and for model evaluation and synthesis are used according to the World Weather Attribution Protocol, described in
Philip et al. (2020a), with supporting details found in van Oldenborgh et al. (2021) and Ciavarella et al. (2021). The essence
of the approach we employ here is that event indices — regional summertime averages of soil moisture, precipitation,
temperature — are represented with continuous probability distributions conditional on GMST, which enables us to estimate
how the event magnitude and probability of occurrence have changed under human-induced climate change. The analysis
steps include: (i) trend calculation from observations; (ii) model validation; (iii) multi-method multi-model attribution and
(iv) synthesis of the attribution statement. We calculate the return periods, the probability ratio (PR — the factor-change in
the event's probability) and change in intensity of the drought event in order to compare the climate of now and the climate
of the past, defined respectively by the GMST values of now and of the preindustrial past (1850–1900, based on the Global
Warming Index https://www.globalwarmingindex.org).

To statistically model the event, we approximate the variable of interest — e.g., soil moisture — by a Gaussian distribution
that incorporates a dependency on global warming. For soil moisture and precipitation, we model the mean and scale
parameters as exponential functions of GMST (for details see Kew et al., 2021), whereas for temperature, the mean
parameter depends linearly on GMST (details in Philip et al., 2020b), which is in line with other research (Wartenburger et
al., 2017, Seneviratne & Hauser, 2020). As such, we use a Gaussian distribution that scales (soil moisture, precipitation) or
shifts (temperature) with GMST, and note that all climate variables of interest are reasonably Gaussian distributed, as one





would expect when examining large regions and seasonal averages (e.g., Schär et al., 2004, Wang et al., 2019). Finally, results from observations and models that pass the validation tests are synthesised into a single attribution statement.

To facilitate comparisons between different models and the observation-driven products, all soil moisture data were scaled prior to the statistical analysis by dividing through the respective 1950–2022 June–August standard deviation.

## 3.4 Model evaluation

Because observation-driven soil moisture products feature large uncertainties owing to the different land surface models employed and their inherent deficiencies (Gevaert et al., 2018), as well as the limitations of remote sensing particularly for the root-zone soil moisture (Babaeian et al., 2019), we rely on precipitation and temperature as proxies for moisture supply and demand in our model evaluation. Rather than directly evaluating the statistical parameters for soil moisture, we require all models to pass validation for the respective domain (WCE, NHET) for both precipitation and temperature. For these variables, we assess the models' fitness for purpose in three ways. First, we qualitatively compare the seasonal cycles in models to observations, checking for the timing and relative amplitudes of peaks and troughs. Second, we compare the spatial pattern of mean summer temperatures for both regions. Third, we check if the parameters of the fitted statistical distribution (Gaussian shifting with GMST for temperature, Gaussian scaling with GMST for precipitation) in models are compatible with those from observation-based estimates. For the observational parameter range, wherever applicable, all of the respective listed observation-based datasets are considered. Models whose statistical parameter range lies within the observational range (95% confidence interval) are considered as 'good', whereas overlapping ranges are 'reasonable'. Additionally, wherever available, the seasonal cycle and spatial pattern of soil moisture were also evaluated against ERA5-Land estimates — these were typically found to be 'reasonable' in the models that passed the combined precipitation and temperature validation. **Suppl. Tables 1 & 2** show the model evaluation results for the root zone soil moisture in the WCE and NHET region, respectively, whereas **Suppl. Tables 3 & 4** present the results for the same regions and surface soil moisture. Only models with an overall performance of 'reasonable' or better were used for the attribution analysis. Based on the capability of the model to capture the seasonal cycle, spatial pattern and statistical properties for temperature and precipitation, a model must pass at least 6 checks, or 8 for models with soil moisture available for evaluation, such as the CMIP6 models, each of which without a single 'bad' performance.

## 3.5 Synthesis

All synthesis figures presented in this study show the changes in probability (a) and intensity (b) of the variable of interest (soil moisture, temperature, precipitation) for the observation-based products (blue) and models (red), and follow the standard analysis method employed by the World Weather Attribution (Philip et al., 2020a). To combine the two lines of evidence into a synthesised assessment, first, a representation error is added (in quadrature) to the observations, to account for the difference between observations-based datasets that cannot be explained by natural variability (light blue bars). This



is shown in the synthesis figures as white boxes around the light blue bars. The dark blue bar shows the average over the observation-based products (black marker) and the total uncertainty (width of the bar) based on natural variability and representation errors. Instead of representation errors, next, a term to account for intermodel spread is added (in quadrature)

to the natural variability of the models. This is shown in the synthesis figures as white boxes around the light red bars. The dark red bar surrounding the model average (black marker) is based on a weighted mean using the respective uncorrelated uncertainties due to natural variability plus intermodel spread. Specifically, the weights are given by the inverse sum of the squared model variability (i.e., the square of the light red bars) and the squared intermodel spread (i.e., the square of the white bars).

Observation-based products and models are combined into a single result in two ways. Firstly, we neglect common model uncertainties beyond the intermodel spread that is depicted by the model average, and compute the average of models and observations using the total respective uncertainties as weights (widths of dark red and blue bars). The resulting weighted average is indicated by the magenta bar. Due to common model uncertainties, the true model uncertainty can be larger than indicated by the intermodel spread. Therefore, we also show the more conservative estimate of an unweighted, direct average

of observations (dark red bar) and models (dark blue bar) contributing 50% each, indicated by the white box around the magenta bar in the synthesis figures. Note that as to not distort the synthesis, we limit very high probability ratios to 10'000.

## 4 Observation-based analysis

### 4.1 Comparing soil moisture across several datasets

The summer of 2022 featured root-zone soil moisture deficits across much of the northern extratropics (**Fig. 1a**). We begin

our analysis by examining regionally averaged July–August soil moisture for NHET (**Fig. 1b**), which is remarkably similar in the last 2 decades with a consistent downward trend for all main datasets (described in Sect. 3.1.1). In the 20th century, the correspondence between different soil moisture estimates is clearly worse, and both ERA5 and ERA5-Land indicate an upward trend whereas GLDAS-CLSM already features a downward trend. This disagreement is most likely a consequence of observation density generally increasing in time (e.g., Dorigo et al., 2015), and the limited availability of satellite data,

especially prior to 1979 (e.g., Dorigo et al., 2012). Nevertheless, all datasets used here indicate that the summer of 2022 featured pronounced — yet not unprecedented — soil moisture deficits averaged across the northern extratropics. Zooming into West-Central Europe (delineated in **Fig. 1a**), we find a good correspondence across all datasets except for the first few decades, providing strong evidence for declining root-zone soil moisture since about 1980 (**Fig. 1c**). Such downward trends have also been noted in other studies (e.g., Trnka et al., 2015, Scherrer et al., 2022). Overall, the 2022 summer drought

signal is stronger in West-Central Europe than in the larger domain, with ERA5, ERA5-Land and GLDAS-CLSM pointing to the driest regionally averaged root-zone soil moisture since 1950. EFAS-historical, the hydrological forecasting and monitoring system used by the EDO and restricted to Europe, indicates that only the summer of 2015 was slightly drier than 2022, but is otherwise consistent with the main datasets.



Since the root-zone soil moisture can only be observed through elaborate, sparse and highly heterogeneously distributed in-
situ measurements, we cannot rely on direct observations for our analysis. Surface soil moisture, on the other hand, can be
sensed from space, although there are several caveats such as dense vegetation resulting in canopy rather than soil water
measurements, as well as limited spatiotemporal coverage, although the latter has been improving. Nonetheless, the main
datasets feature largely similar root-zone and surface soil moisture interannual variability and long-term changes. This is
easiest observed when comparing the datasets without subtracting the baseline as done in **Fig. 2**, although the
correspondence of soil moisture between surface layer and root-zone is lower in ERA5 than for GLDAS-CLSM and ERA5-
Land. Nevertheless, the overall temporal evolution of summer soil moisture in the surface layer and root-zone is consistent in
both regions for all main datasets, which is plausible given that soil moisture near the surface and in deeper layers is
inherently connected through infiltration and diffusion processes (e.g., Albergel et al., 2008).

Next, we extend our analysis by comparing surface soil moisture across a total of 7 and 8 products for the northern
extratropics and West-Central Europe, respectively, by adding several supplementary datasets that incorporate either
microwave-sensed or in-situ soil moisture measurements (see also Sect. 3.1.2). In ERA5-Land, the spatial pattern of soil
moisture anomalies is fairly similar for the surface and the root-zone (cf. **Fig. 1a**, **3a**), and this also applies to the other main
soil moisture datasets. For the mean surface soil moisture across the northern extratropics (**Fig. 3b**), almost all datasets agree
on an overall decline in the last two decades. Only the satellite-based, gap-free RSSSM shows an upward trend for 2003–
2018, as already noted by its creators (Chen et al., 2020). The downward trend is also observed for the gap-filled products
from ESA-CCI and CLIMFILL, which show a remarkable correspondence even though the original, not gap-filled soil
moisture from ESA-CCI features upward trends for the northern extratropics, especially prior to 2008 (not shown). None of
the supplementary datasets available for NHET covers the year 2022, but ESA-CCI and CLIMFILL do not seem to exhibit
the clear decline evident for the ECMWF products and GLDAS-CLSM after 2017. Otherwise, both of these remote-sensed
estimates show a particularly good agreement with GLDAS-CLSM, whereas SoMo.ml, a machine-learning based product
relying on in-situ soil moisture and ERA5 meteorological forcing, is fairly consistent with ERA5 but portrays a slightly
weaker downward tendency.

While there seems to be some inconsistency with regards to long–term changes in NHET soil water content, we are not
aware of any recent studies that have discussed positive northern hemispheric or global soil moisture trends. A tendency
toward drying — especially for the surface and during summer — has been reported in several analyses (e.g., Sheffield &
Wood, 2012; Cheng & Huang, 2016; Deng et al., 2020; Qiao et al., 2021). This increases our confidence in the two selected
soil moisture datasets for further analyses, GLDAS-CLSM and ERA5-Land, as both feature downward trends for the surface
and for the root-zone soil moisture. We cannot reliably assess, however, whether these products are truly more accurate than,
e.g., RSSSM, which features a recent surface soil moisture increase at the hemispheric or global scale.



For West-Central Europe, on the other hand, none of the 8 available surface soil moisture products indicates clear upward
trends in the ongoing millennium, and the overall agreement between the different estimates is better than for the northern
extratropics (**Fig. 3c**). We remark that, based on the last two decades, the remote-sensed products show smaller drying
tendencies than the other datasets used here, but also point out that the short available time period complicates such
assessments. Moreover, this domain is much more observationally constrained than the entire northern extratropics, and in
particular the high latitudes, and we hence deem the choice of soil moisture dataset for the attribution analysis less critical
than for the larger domain.

    Finally, we also briefly inspect the total water storage as measured by the Gravity Recovery and Climate Experiment
(GRACE). These data are not only measured in a fundamentally different manner than remote-sensed surface soil moisture,
but also represent the sum of all above and below surface water storages (e.g., canopy water, rivers and lakes, groundwater,
and, of course, soil moisture) and hence typically serve as a proxy for groundwater drought. As such, the downward trends
evident for both domains (**Suppl. Fig. 1**), which is especially pronounced for West-Central Europe, cannot directly validate
long-term changes in soil moisture. Nevertheless, the observed drying tendency is fully consistent with the declining root-
zone soil moisture in the last two decades evident in **Fig. 1**. Overall, we conclude that the comparison to supplementary
datasets strengthens our analysis, but also emphasise that the observation-based attribution of the 2022 soil drought to
human-induced climate change may be associated with more uncertainty than represented by GLDAS-CLSM and ERA5-
Land alone, particularly for the northern extratropics.

## 4.2 Event return period and long-term trend analysis

    In a next step, we investigate the probability of the 2022 soil drought as well as the anthropogenic fingerprint for both
analysis regions. Whereas the previous Sect. presented soil moisture as a function of time, here, we explore the relationship
between the warming climate and soil moisture.

### 4.2.1 West-Central Europe

    We fit June–August root-zone soil moisture averaged over the WCE region as a function of GMST, as described in Philip et
al. (2020a), for ERA5-Land (**Fig. 4a**) and GLDAS-CLSM (**Fig. 4b**). The left panels depict soil moisture as a function of the
GMST anomaly, while the right panels show the corresponding Gaussian distribution-based return period curves in the
present 2022 climate (red lines) and the past, 1.2 °C cooler climate (blue lines). The return periods are 12 and 33 years
according to ERA5-Land and GLDAS-CLSM, respectively. We average and round this to a return period of 20 years for the
remainder of the analysis. Our analysis for ERA5-Land points to probability ratios well above one (95% confidence interval
of 4 .. 450), and estimates based on GLDAS-CLSM are several orders of magnitude higher with a lower bound of 90'000,
suggesting an even stronger warming signal (Table S5). We also estimate the mean change in WCE summer root-zone soil
moisture from the past to the present climate, and obtain best estimates (confidence intervals) of -9% (-13% .. -4%) for





ERA5-Land and -14% (-16% .. -11%) for GLDAS-CLSM. Despite the apparent mismatch of the probability ratios, there is an overlap in confidence intervals of mean intensity changes. The latter are less sensitive than the probability ratios to the inferred relationships between global warming and long-term soil moisture changes, since they inform on the slope of the linear trend rather than a ratio of occurrence probabilities, for which the denominator — the probability of the event for the past climate — can become very small, as is the case for GLDAS-CLSM.


Overall, these results indicate that an event such as the 2022 summer drought in WCE has become far more likely due to our warming climate. We also perform an analogous analysis for the June–August average temperature and precipitation (**Suppl. Figs. 2** & **3**). Temperature shows very strong trends with probability ratios of at least 170 for E-OBS data, and even much larger for ERA5 data. This corresponds to a change in intensity of about 1.7°C to 2 °C (for details see **Suppl. Table 6**). The return period used for the model analysis of temperature in the WCE region is 20 years. Trends in precipitation are much smaller and encompass no change (**Suppl. Table 7**), and we employ a return period for the model analysis of a low precipitation event in the WCE region of 10 years.


### 4.2.2 Northern extratropics

Similarly to WCE, the return period of the 2022 summer root zone soil drought in the northern extratropics ranges from 10 to 20 years (**Fig. 5**), and we also employ a value of 20 years for the subsequent model analyses. ERA5-Land-based data gives a probability ratio of around 700 (50 to 70,000) and GLDAS-CLSM even larger, with a lower bound already on the order of 10 Mio. The corresponding changes in intensity of root-zone soil moisture are -2.4% (-3.2% to -1.5%) for ERA5-Land, and -3.1% (-3.6% to -2.7%) for GLDAS-CLSM. Compared to the much smaller European region, there is thus a weaker tendency toward soil drying in summer, yet the warming-induced change in probability of occurrence of a 2002-like soil moisture deficit is even higher in NHET. We attribute this to the fact that the interannual variability of climate variables tends to decrease at larger spatial scales, especially for precipitation but also for temperature (Giorgi, 2002, Lehner et al., 2020), so that the anthropogenic signal emerges more clearly for the northern extratropics despite weaker downward soil moisture trends. We complement our investigation by analysing the June–August average temperature and precipitation for the NHET domain (**Suppl. Figs. 4 & 5**), with temperature showing strong trends and excessive probability ratios for ERA5 data. This indicates that such a hot summer would have been virtually impossible without climate change, and the corresponding change in intensity is about 1.9 ºC with a 95% confidence interval of 1.7 ºC to 2.1 ºC (for details see **Suppl. Table 9**). The return period used for the model analysis of temperature in the NHET region is 10 years. As for the WCE region, the trend in precipitation is much smaller and encompasses no change, see **Suppl. Table 10** for details. The return period used for the model analysis of a low precipitation event in the NHET region is 10 years.







## 5 Hazard synthesis using observation-based datasets and models

In a final step, we combine results from observations-based products — the offline reanalysis or observation-driven land surface model simulations ERA5-Land and GLDAS-CLSM — and models that passed the evaluation. This synthesis,
explained in Sect. 3.5, enables us to give overarching attribution statements building on all the employed simulations and observation-driven estimates.

### 5.1 West-Central Europe root-zone soil moisture

For probability ratios of WCE, **Fig. 6** reveals large representation errors (white bars surrounding observational estimates), owing to the fact that the confidence intervals of observation-based estimates (light blue shading) do not overlap. The model
uncertainty is comparatively low, and the probability ratio averaged across models of 2.2 (0.4 to 13) is notably lower than for the observation-based estimates with 546 (0.1 to 2.3*10^6). When combining models and 'observations' according to their visualised uncertainties, the high representation error results in a synthesis dominated by the models, with a probability ratio best estimate of 2.8 (0.5 to 16). This partly holds for the change in intensity as well, for which the models also show a weaker signal than the observation-driven soil moisture products, synthesised to a best estimate of -3.7% (-7.4 to 0.1 %). We
point out that here, consistent with the World Weather Attribution Protocol (Philip et al. 2020a, van Oldenborgh et al., 2021), we rely on historical climate simulations  extended with one of the climate scenarios up to the event year, 2022. This makes the statistical analysis more robust due to the larger sample size, from 1850 onwards, compared to the observation-driven estimates.

Nonetheless, we repeat this analysis in the next step, enforcing a uniform analysis period of 1950–2022 for all datasets and
models to use a consistent time period for models and observations. The resulting synthesis plot (**Fig. 7**) is the product of the same methodological steps used to create **Fig. 6**. Given that the long-term evolution of soil moisture is dominated by global warming in the models, these figures should depict similar best estimates since our analysis evaluates soil moisture changes as a function of warming rather than time. Compared to recent decades since about the mid-1980s, when the period of global dimming had ended (Wild et al., 2005), there was little warming between 1850 and 1950, and soil moisture is expected to
portray at most weak trends. For some models, such as CESM2-WACCM, WCE root-zone soil moisture is fully consistent with this expectation, depicting a moderate decline up until 1980, followed by sharp decrease (**Fig. S6**). Other models, such as, e.g., MPI-ESM1-2-HR, feature increasing soil moisture from pre-industrial times into the second half of the 20th century for the same region, followed by clear downward trends. In other words, for some models, the sign of the apparent relationship between GMST and soil moisture changes, which is consistent with a study pointing to potentially non-linear
scaling of soil moisture with global warming (Lehner & Coats, 2021), and can mask the emerging response to strong global warming in recent decades within our (linear) statistical framework. This is why when we restrict the analysis period to 1950 onwards, **Fig. 7** depicts a probability ratio on the order of 10 for MPI-ESM1-2-HR, whereas it is < 1 in **Fig. 6**. For CESM2-





WACCM, on the other hand, whose long-term root-zone soil moisture in WCE evolves in line with the nonlinear GMST increase, the probability ratio remains between 4 and 5 for both analysis periods.

For most models, **Fig. 7** shows both higher probability ratios as well as stronger mean soil moisture declines than **Fig. 6,** with a synthesised probability ratio of 8.8 (0.8 to 93.6) and changes in intensity of -7.5% (-12.5% to -2.1%.). As a consequence, overall, the model probability ratios are fairly consistent with ERA5-Land, although GLDAS-CLSM still features a much stronger warming signal. In terms of the more robust changes in intensity, GLDAS-CLSM is fairly similar to MPI-ESM1-2-LR, the model with the strongest signal, whereas ERA5-Land is closer to the remaining models. As outlined

in Sect. 4.1, the different observation-driven datasets agree on the decline of WCE root-zone soil moisture after 1980, yet the ECMWF products suggest an upward tendency prior to 1980 that is largely absent in GLDAS-CLSM. Such disagreements are also found among the models; e.g., MPI-ESM1-2-LR suggests that WCE root-zone soil moisture decreases notably sooner than MPI-ESM1-2-HR, and is similar to GLDAS-CLSM with downward trends since about 1960. GLDAS-CLSM and MPI-ESM1-2-LR likely indicate the strongest anthropogenic fingerprint in root-zone soil moisture precisely because of

this, as our statistical approach can infer a stronger link between global warming and soil moisture changes. Finally, we remark that among the 25 available CMIP6 models used here (of which 7 passed the validation), all agree that based on 1950–2022, the best estimate of the probability ratio is at least 1, and oftentimes on the order of 10 or higher. Nonetheless, the lower bounds of the probability ratio for all validated models and observation-based estimates — with MPI-ESM1-2-LR and GLDAS-CLSM being the sole exceptions — are below 1, while the corresponding upper bound of the change in

intensity is positive, suggesting that a weaker or even opposing response to global warming than suggested by the best estimates is possible, albeit unlikely. Overall, our analysis indicates a human-induced summer root-zone soil moisture decline in West-Central Europe, rendering the 2022 soil drought more likely than in a pre-industrial climate, although the associated uncertainties are high.

### 5.2 Northern extratropics root-zone soil moisture

Moving on to the northern extratropics, for which far more models have passed validation (always performed for the respective domain), the synthesised probability ratio using the weighted average is much larger than for the WCE region with a probability ratio of 877 (25 to 39,900). The unweighted synthesis, that is, averaged giving equal weight to observation-based estimates (blue bar) and models (red bar), has a similarly large upper bound, whereas the lower bound amounts to 4 (**Fig. 8a**). For such high probability ratios, the exact quantification of the best estimate is highly uncertain,

hence we use the (weighted) lower bound as the synthesised result, which suggests that anthropogenic climate change has increased the likelihood of the NHET root zone soil moisture event by a factor of at least 20. The synthesised change in intensity is -2.5% (-3.7% to -1.3%) when combining individual models and observation-based estimates according to their total uncertainties (**Fig. 8b**), and is similar for an unweighted average. Overall, **Fig. 8** shows that most validated models agree with the employed observational datasets in that an event such as the 2022 summer soil drought has become more



likely under global warming. But while the observation-based products agree that the probability ratio is larger than 1 (and the change of intensity below 0) based on their confidence intervals, a clear majority of the models feature a weaker and less certain warming response. Compared to WCE (cf. Figs. 6 or 7), the intermodel spread is higher, whereas the representation error of the observation-based products is lower. We note again, however, that in light of our findings in Sect. 4.1, GLDAS-CLSM and ERA5-Land may not sufficiently capture the true 'observational' uncertainty, and hence we emphasise here that

these root-zone soil moisture attribution results, particularly for the NHET region, should be interpreted with caution.

## 5.3 Temperature and precipitation

The 2022 summer was characterised by unusually hot and dry conditions in West-Central Europe and across much of the northern extratropics, as evidenced by temperature surpluses and precipitation deficits. For WCE and compared to the entire analysis period (1950–2022), we obtain standardised precipitation anomalies of -1.4 and -2 according to ERA5 and E-OBS,

whereas temperature anomalies amount to 2.3 in both products. ERA5 features similar anomalies in the northern extratropics, with -1.3 and 2.2 for precipitation and temperature, respectively. Considering these pronounced anomalies, excessive heat and precipitation shortages likely played an important role in the occurrence of soil drought in the 2022 summer. As noted in Sect. 4.2.1 & 4.2.2, however, we have found clear upward temperature trends for temperature yet no clear precipitation changes for the two regions in ERA5 (and E-OBS for WCE). Here, we also include model results to

further examine changes in precipitation and temperature (**Suppl. Figs. 7–10**). Using the same synthesis procedure, the weighted average for temperature in the WCE region is PR = 2430 (214–26400), with a change in intensity of 1.8 (1.1 to 2.5) °C, (**Suppl. Fig. 7**). For the northern extratropics, the change in intensity is 1.8 (1.4 to 2.2) °C, and thus very similar to the result for WCE. The synthesised probability ratio, on the other hand, is even higher than for WCE, so high that we refrain from a quantification, and instead limit the values to 10'000 (see also Sect. 3.5), thereby confirming the finding from the

observational analysis that the extreme temperatures over the NHET region would have been virtually impossible without climate change (**Suppl. Fig. 9**). This is consistent with our soil moisture analysis, for which a stronger warming signal emerged in the larger region. In contrast, the change in precipitation is centred around 1 for both regions (**Suppl. Figs. 8 & 10**), with no clear changes in intensity.

These results suggest that for both domains, trends in root-zone soil moisture are likely fueled by increasing temperatures,

since no clear signal emerges for precipitation. Previous research by Cheng & Huang (2016), who argue that the interannual to decadal variability of soil moisture tends to be controlled by precipitation whereas long-term changes are dominated by upward temperature trends, is consistent with our findings. This does not imply that precipitation shortages were irrelevant for the occurrence of soil drought in the 2022 summer, but rather that these rainfall deficits are primarily manifestations of natural variability. The regional summer temperatures are, of course, also subject to natural variability, but additionally

reveal a clear warming signal that considerably boosts the probability of occurrence of marked positive anomalies such as in 2022.



### 5.4 Surface soil moisture

To complement our statistical analysis of the relationship between the warming climate and root-zone soil moisture in West-Central Europe and the northern extratropics, we also attribute the 2022 surface soil moisture drought. Since long-term changes in observation-based root-zone and surface soil moisture estimates seem largely consistent for both domains (**Fig. 2**), we expect similar results as for the analysis of agro-ecological drought. ERA5 — which, unlike ERA5-Land, assimilates soil moisture data from scatterometers and the global SYNOP network — is also considered for the analysis of surface soil moisture, as the discontinuities due to the use of multiple production streams for the reanalysis only affect deeper soil layers.

We provide the event return period and long-term trend analysis for surface soil moisture in the Appendix, and proceed with the same return period of 20 years as for root-zone soil moisture. For the surface soil moisture in West-Central Europe, the synthesised probability ratio using the weighted average is 5.9 (0.71 to 50); whereas the unweighted upper bound is much larger at 1150, the lower bound in this case is similar to the weighted average, as shown in **Suppl. Fig. 11a**. As for WCE root-zone soil moisture, we use the rounded best estimate as the synthesised result, suggesting anthropogenic climate change has increased the likelihood of the WCE surface soil moisture event by a factor of 5–6. The change in intensity for the same event is shown in **Suppl. Fig. 11b**, and averages -8.7% (-14 to -2.7%).

For the surface soil moisture in the northern extratropics, the synthesised probability ratio using the weighted average is again much larger than for the WCE region with a probability ratio of 450 (5 to 44000), as shown in **Suppl. Fig. 12a**. Consistent with our analysis of root-zone soil moisture in the northern extratropics, we rely on the lower bound as the synthesised result; anthropogenic climate change has increased the likelihood of the NHET surface soil moisture event by a factor of at least 5. The change in intensity for the same event is shown in **Fig. S12b**, suggesting an average decrease of 3.3% (-5.7 to -0.85%).

### 5.5 Synthesis for an additional warming of 0.8 °C

We also assessed how the frequency and intensity of the two types of soil moisture drought in both regions would change in a 0.8$o$C warmer world compared to today. For all event definitions, a further increase in intensity as well as a 2–30-fold further increase in the frequency of such an event is found (**Suppl. Figs. 13–16**). In combination with the strong trends in temperature extremes, these results strengthen our confidence in the soil moisture results, even though an exact quantification is difficult due to the difficulties in measuring soil moisture and resulting large discrepancies in observation-based data sets.

### 6 Conclusion

Extending our rapid attribution analysis (Schumacher et al., 2022), we find evidence for a global warming-induced summer root-zone soil moisture decline in West-Central Europe, and several observation-driven soil moisture estimates agree on a





downward trend since at least 1980. Our analysis suggests that the large uncertainties, also due to a lack of in-situ root-zone soil moisture observations except for a few hundred stations, make it difficult to communicate precise numbers, but the synthesised probability ratio for a 2022-like summer drought in West-Central Europe is likely larger than 1 and amounts to

about 5. In other words, combining observation-driven and model evidence, we find that anthropogenic climate change has made such an event more probable. We emphasise here, however, that the lower bound of the synthesised probability ratio is below 1, and hence we cannot exclude the possibility that the likelihood of an event such as the 2022 soil drought has not been modulated (or even decreased) by human-induced global warming. This also applies to the northern extratropics, where our analysis suggests a stronger overall warming signal — with a probability ratio of at least 20  — and associated decline in

root-zone soil moisture. The "observational" uncertainty is higher than for West-Central Europe, however, and hence this result should be treated with caution. Moreover, observation-based soil moisture products do not agree on when the warming signal becomes evident, with GLDAS-CLSM displaying drying tendencies about 2 decades earlier than the ECMWF products for both regions. Similarly, nearly all CMIP6 models display declining summertime root-zone soil moisture throughout the 21st century in West-Central Europe and the entire northern extratropics, but there is no agreement whether

this decline started in recent decades or already in pre-industrial times. This could indicate that long-term soil moisture changes are not solely driven by global warming, and hence only emerge clearly in the presence of strong warming. In snow-dominated regions it is also possible that changes in snowpack and precipitation partitioning in winter and spring influence soil moisture droughts in the subsequent summer (Wieder et al., 2022).

Nonetheless, and in line with previous research, our results point to increasing temperatures as a key driver behind declining

soil moisture in West-Central Europe and across much of the northern extratropics. Our analysis of surface soil moisture provides additional evidence for an enhanced tendency toward soil drought in both regions, with similar results as obtained for the root-zone. According to the reanalysis and observation-driven land surface models ERA5-Land and GLDAS-CLSM, low summer soil moisture such as observed in 2022 happens about once in 20 years in today's climate in both regions. For a pre-industrial climate (1.2°C cooler than the present), a similarly intense soil drought would take place in West-Central

Europe roughly once per century, and even less often in the northern extratropics. In this context, we point out that our analysis has been largely restricted to the lower bounds and best estimates of the synthesised probability ratios and intensity changes. This appears adequate considering that the uncertainty in the attribution of extremes in soil moisture is higher than for variables such as temperature, and hence we intentionally stay on the conservative side. Even so, in light of the high upper bounds, we also mention the possibility that our best estimates underestimate the decline in soil moisture in response

to a warming climate, in which case widespread drought conditions as in the 2022 summer would have been virtually impossible without human-induced climate change. Moreover, the models analysed also show that soil moisture drought will continue to increase with additional global warming — in West-Central Europe, a 2022-like event or worse is expected to occur about every 10 years once a warming level of 2°C is reached, and nearly every single year in the northern extratropics.





This is consistent with projected long-term trends in climate models as reported, e.g., in the IPCC AR6 (IPCC, 2021), and

670 should serve as a strong motivation to increase our efforts to limit future global warming.










**Code availability**

The code used to process the data and perform analysis can be obtained from the corresponding author upon request.

**Data availability**

The data used for the statistical analyses are available via the Climate Explorer (https://climexp.knmi.nl/start.cgi).

**Author contribution**

HKB, SL, WY, MHau, MHir, VB, and DLS prepared data for analysis and/or contributed post-processing code; MZ, FO, CB, SP, SK, SL, WY, and DLS analysed the data; MZ, FO, CB, SP, SK, MV, RS, DH, JA, MvA, LJH and DLS wrote the first manuscript draft, and all authors reviewed and edited the manuscript.

**Competing interests**

The authors declare that they have no conflict of interest.

**Acknowledgements**

This analysis was partly funded by the XAIDA H2020 project, and has received funding from the European Union's Horizon 2020 research and innovation programme under grant agreement No. 101003469.






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

1090

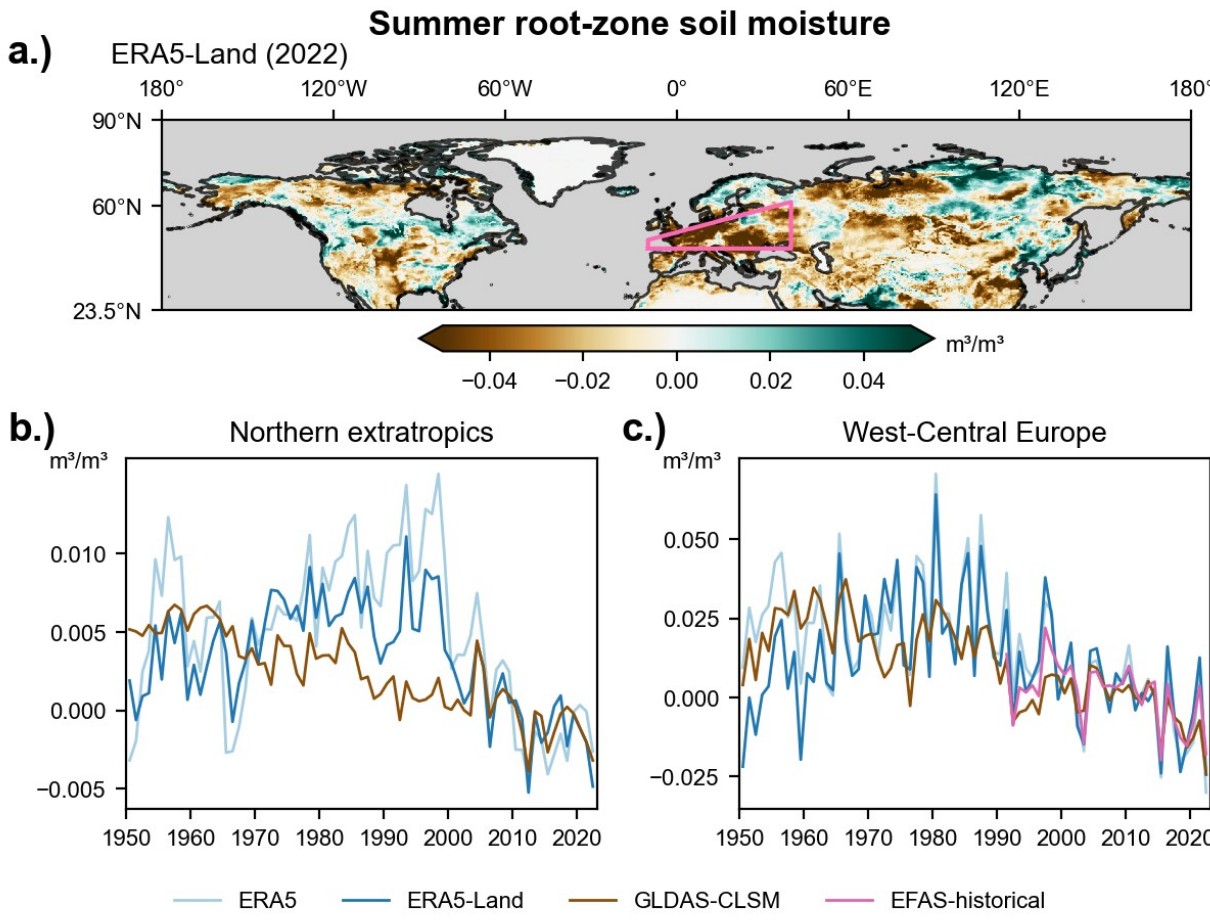

**Figure 1:** (a) Mean summer (June–August) root-zone soil moisture in 2022 over the northern extratropics (NHET), shown for the ERA5-Land dataset and expressed as anomalies with respect to 1950–2022. West-Central Europe (WCE) is highlighted by the pink contour. (b) Summer root-zone soil moisture averaged over the northern extratropics for the main datasets used for analysis, with the 2003–2018 baseline subtracted to facilitate the comparison. (c) Like (b), but for West-Central Europe. Note that the supplementary dataset EFAS-historical is also shown, but this product is only available for Europe and hence not used for (b), and that the second ("upper") soil layer — which does not represent a fixed depth, unlike for the other datasets displayed here (1m) — is selected to represent the root-zone.



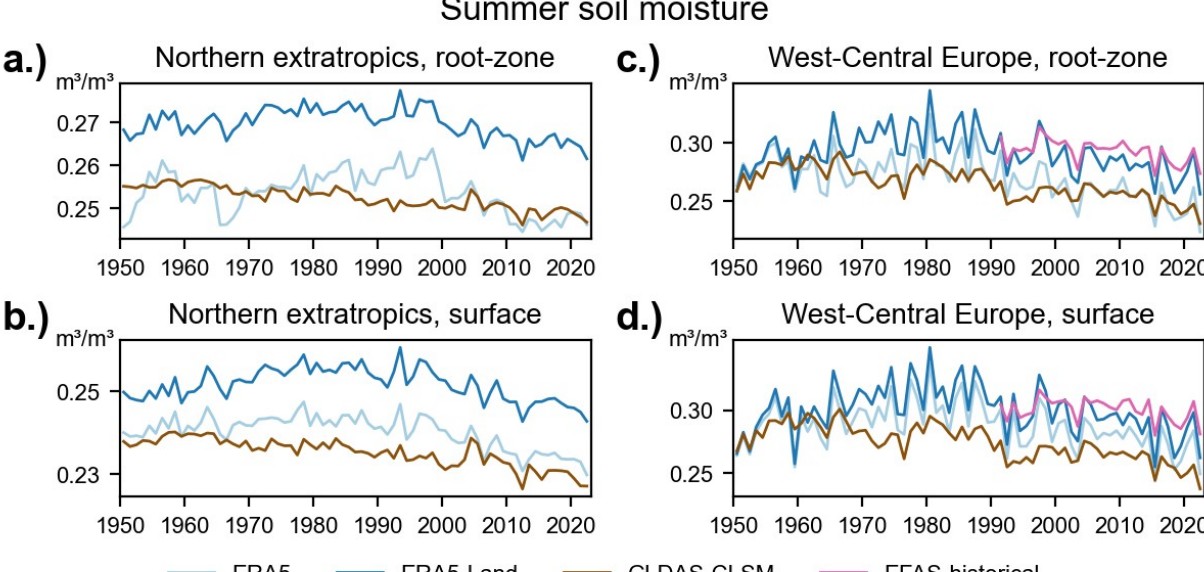

**Figure 2:** Summer soil moisture averaged over the northern extratropics and West-Central Europe for the same datasets as in

**Fig. 1**, but for both the root-zone — a.), c.) — and surface layer — b.), d.) — for the two domains. No baseline is subtracted

here for a convenient comparison of individual products across the different soil depths.



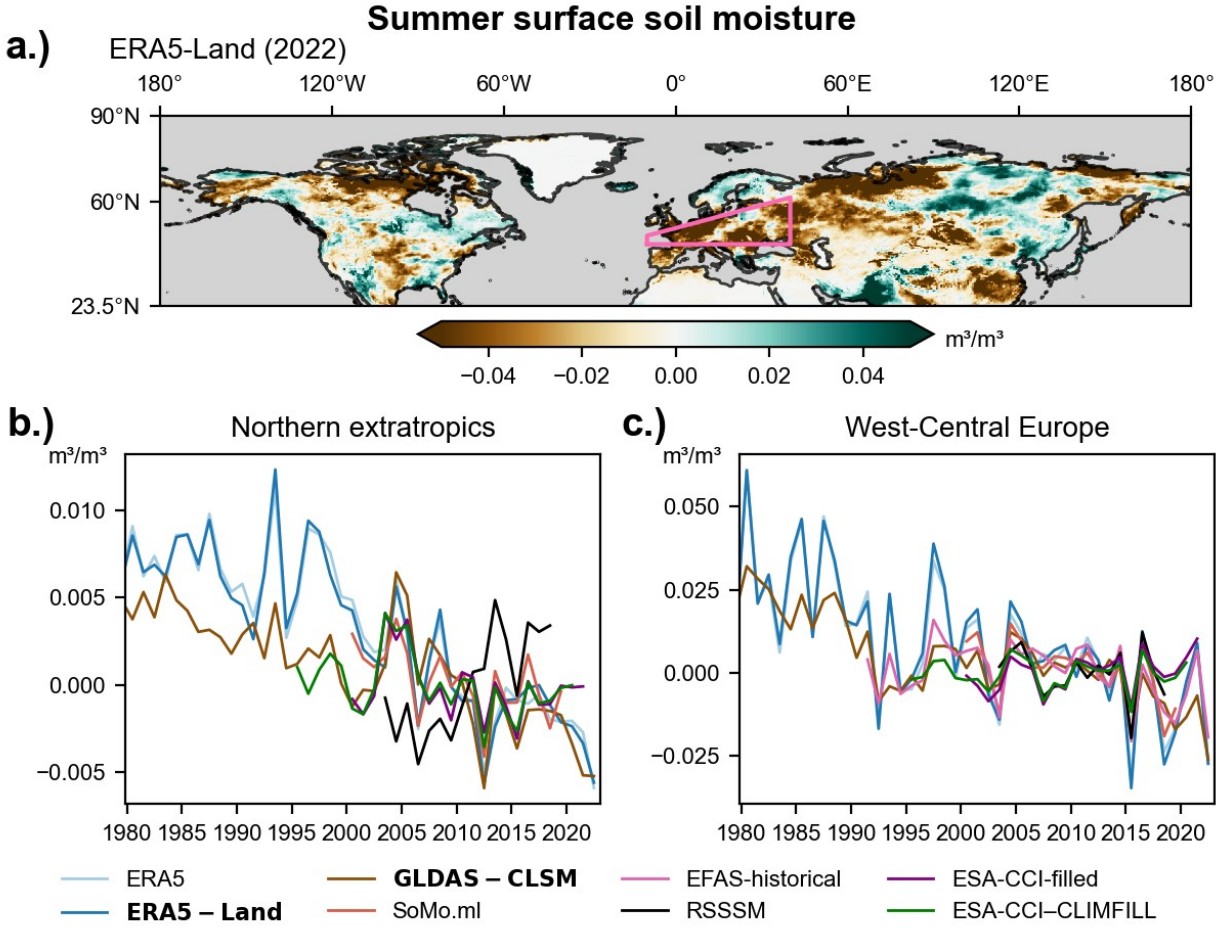

**Figure 3: (a)** Summer average surface soil moisture in 2022 over the northern extratropics (NHET), shown for the ERA5-Land dataset and expressed as anomalies with respect to 1950–2022. West-Central Europe (WCE) is highlighted by the pink box. **(b)** Summer surface soil moisture averaged over the northern extratropics for several datasets, with the 2003–2018 baseline subtracted to facilitate the comparison. **(c)** Like (b), but for West-Central Europe. Note that for EFAS-historical, the first ("superficial") soil layer — which does not represent a fixed depth, contrary to the other datasets shown here — is selected to represent the surface soil moisture. The two main datasets employed for both the root-zone and surface soil moisture event attribution are highlighted in the legend (bold font).





**(a)** based on ERA5-Land



**(b)** based on GLDAS-CLSM

**Figure 4: Summer WCE root-zone soil moisture under global warming.** Gaussian fit with location parameter scaling proportional to GMST and constant dispersion parameter, for the WCE region and based on (a) ERA5-Land and (b) GLDAS-CLSM. The 2022 event is included in the fit. **Left**: Observed summer mean surface soil moisture as a function of the smoothed GMST. The thick red line denotes the time-varying location parameter. The vertical red lines show the 95% confidence interval for the location parameter, for the current, 2022 climate and a 1.2ºC cooler climate. The 2022 observation is highlighted with the magenta box. **Right**: Return time plots for the climate of 2022 (red) and a climate with GMST 1.2 ºC cooler (blue). The observations are shown twice: once shifted up to the current climate and once shifted down to the climate of the late nineteenth century. The markers show the data and the lines show the fits and uncertainty from the bootstrap. The magenta line shows the magnitude of the 2022 event analysed here.





**Figure 5: Summer NHET root-zone soil moisture under global warming.** As Fig. 4, but for the northern extratropics.

1135



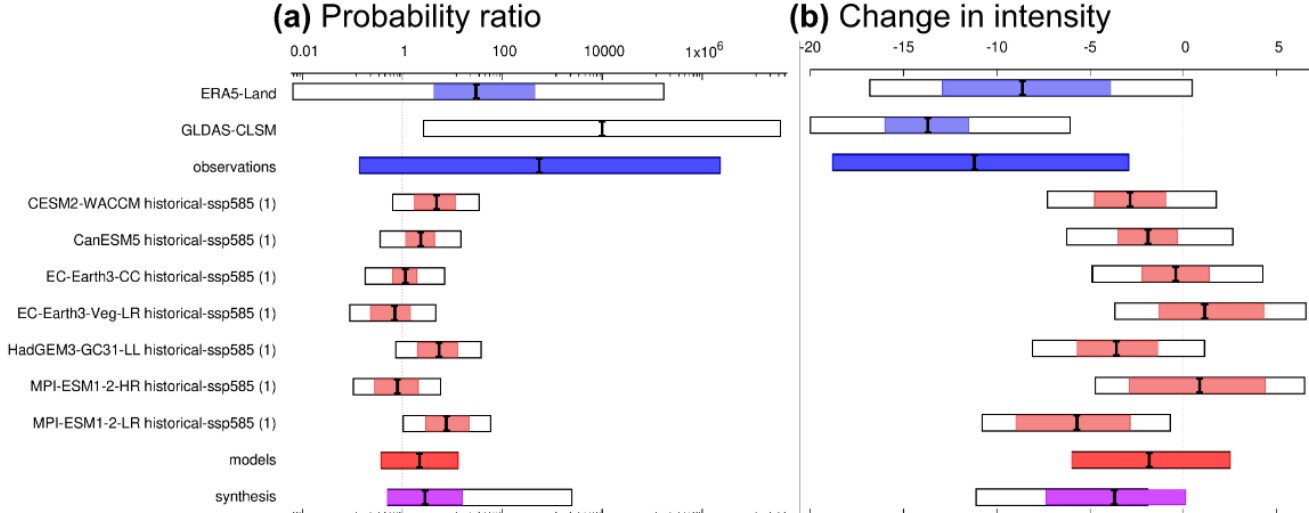

**Figure 6: Synthesis for WCE root-zone soil moisture.** Synthesised (a) probability ratios and (b) intensity changes (%) when comparing the return period and magnitudes of the 2022 summer root zone soil moisture for the WCE region in the current climate and a 1.2oC cooler climate. Note that while the employed observation-based products are restricted to 1950–2022, for models, we make use of the additional available data for the statistical analysis (1850–2022).

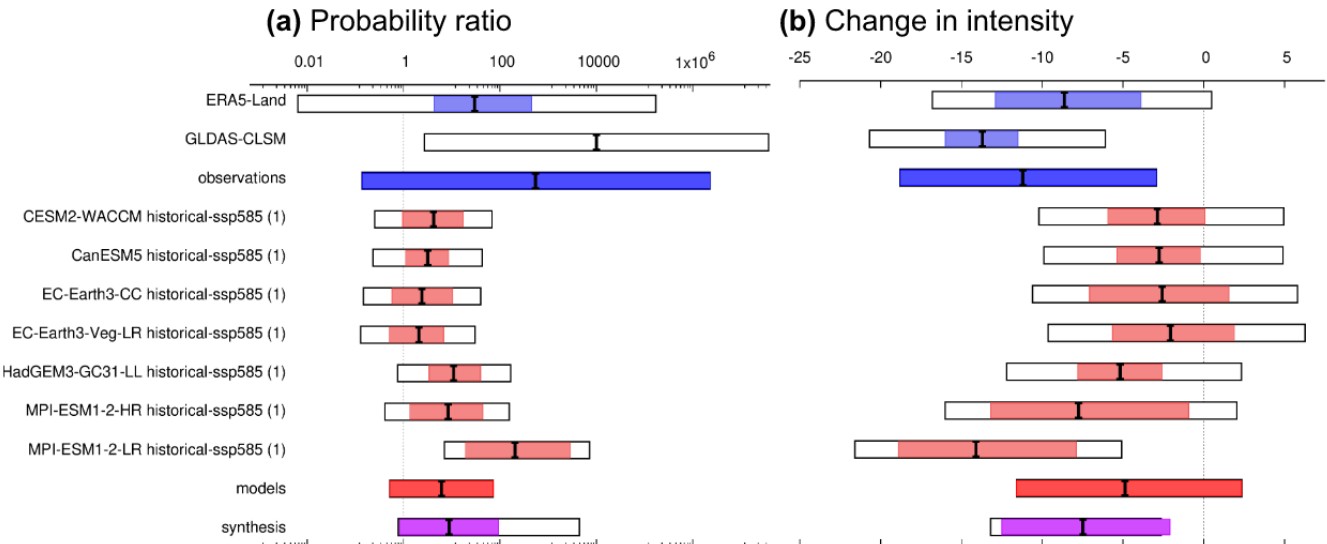

**Figure 7:** As Fig. 6, but restricting the model data to 1950–2022, and hence using consistent time periods among observation-driven estimates and models.





**Figure 8: Synthesis for NHET root-zone soil moisture.** Synthesised (a) probability ratios and (b) intensity changes (%) when comparing the return period and magnitudes of the 2022 June–August root zone soil moisture for the northern extratropics in the current climate and a 1.2 oC cooler climate. Note that while the employed observation-based products are restricted to 1950–2022, for models, we make use of the additional available data for the statistical analysis (1850–2022).