# Peer review of "Detecting the human fingerprint in the summer 2022 West-Central European soil drought"

_EGUsphere, 2023_

## Author Comment (AC1)

**REVIEW #1**

General comments

This work presents novel and interesting results that will be of interest to the readers of this journal and the manuscript is well written. The authors have presented a well-informed investigation and conducted a thorough examination of relevant datasets. I can find no major problems. There are a number of issues and corrections that should be addressed before publications though I consider these minor and that the work is substantially complete.

The work is an event attribution study of the 2022 West-Central European soil drought, focussing on types of drought relevant to ecology and agriculture. The authors perform a detailed review of the state of the science including an assessment of impacts and the work is well targeted in this sense. The chosen methodology is that of the World Weather Attribution group, designed for rapid attribution and hence highly prescriptive but also by now applied in publication a number of times. This study is complex, providing analyses of a large number of different models of various types, different estimates of observed soil moisture, all over two different regions and with supporting analyses of temperature and precipitation. While clarification on points of methodology is sometimes required I judge that the application of the methodology has been appropriate, successful and useful. The authors are at pains to emphasise the sources of uncertainty in the analysis, not least of which arise from the observations or observations-based products. This uncertainty is explored at length and conclusions have been well phrased in light of this. The authors provide a convincing argument that robust yet conservative conclusions about the change in soil drought can be made and highlight that with lower confidence stronger statements are possible.

We thank the reviewer for their positive and thorough assessment of our work, and reply individually to all specific questions below using blue font.

Specific questions

1. L313 "three multi-model ensembles using different framings". These models are all used as part of a single multi-model ensemble and (as far as I can see) are treated equally. As this involves a parametric fit to all of the residuals to the

same covariate then the framing referred to appears to be the same for every model. If I'm right then the statement would better be along the lines of "three multi-model ensembles of different types of models that are combined into a single multi-model ensemble treated under the same framing."

We agree that the reviewer's suggestion more accurately describes our approach, which indeed relies on three multi-model ensembles consisting of different model types that are combined into a single ensemble after the model evaluation. We have edited the text as follows:

We use several climate modelling experiments in this study, building on  three multi-model ensembles of  different  model types (Philip et al., 2020a): coupled global circulation models (GCMs), high-resolution models, and sea surface temperature (SST) driven GCMs. All models are evaluated, and the simulations of models that pass the required checks (Sect. 3.4) are combined into a single multi-model ensemble that is subsequently treated under the same framing.

2. L316 Use of SSP5-8.5 scenario. Given that use of this scenario can be criticised for certain uses, it might be worth a comment that as the future scenario is based on a fixed warming level (2°C) not tied to timing of when that level is reached, then the use of this scenario is defensible.

Thank you for this clarifying suggestion. We believe it fits best in Section 3.3, where the statistical approach is described, rather than where the different model ensembles are introduced. The text has been extended and edited as follows:

We note that, regardless of the underlying emission scenario, model data from 1850 to 2022 and from 1850 to 2050 are used to conduct the present-vs-past and future-vs-present climate analyses, but these time periods only indicate the amount of data used to fit the statistical model and hence infer the relationship between event indices and GMSTs. We then rely on global warming levels to calculate the return periods, the probability ratio (PR — the factor-change in the event's probability) and change in intensity of the drought event. . For our comparison of the present (2022) to the past (1850–1900) climate, the GMST changes with respect to the present amount to -1.2 ° C according to the Global Warming Index (https://www.globalwarmingindex.org), and for comparing additional changes in the future to the present, we use +0.8 ° C relative to the 2022 GMSTs (+2.0 ° C with respect to pre-industrial conditions). In other words, we are conditioning the analysis on observed and simulated global warming levels and not on specific time frames. As such, it does

not matter when the future warming is reached in any given model simulation, which allows us to combine models with different emission scenarios and still perform a consistent analysis.

3. L324 So is the AMIP model over-represented by 10x compared to the CMIP6 models (which use 1 member each) or is there an inverse ensemble size weight applied to each model?

Following the World Weather Attribution protocol, multiple ensemble members are pooled together for the statistical analysis, such that the synthesis is based on different estimates per model, regardless of the underlying number of ensemble members. In this sense, no ensemble size weighting is applied, and models with several ensemble members are not over-represented. The following sentence has been added to Sect. 3.3:

Where applicable (see Sect. 3.3), multiple initial condition ensemble members are pooled together for the statistical evaluation and analysis (e.g., the 10-member AMIP AM2.5C360 ensemble).

4. L348 Use of GWI. Obviously other estimate of the change in GMST attributable to human influence are available, though I don't expect this to make a substantial difference to the conclusions. Has GWI been used for this purpose in another study using the WWA protocol?

The Global Warming Index (GWI) is only used to estimate the current warming level compared to the pre-industrial baseline. This warming level is then used to calculate the return periods and changes in event probability and magnitude in the present climate compared to the preindustrial past using the already calibrated statistical models. The covariate used to fit the models, GMST, is taken from GISTEMP in the case of observations, or the respective model. We note that GISTEMP, processed as described in the manuscript (with a 4-year moving average), is 1.15 °C warmer in 2022 than 1879–1900, which is close to the 1.2 °C we employ with respect to 1850-1900. Whereas employing slightly lower or higher GMST changes than 1.2 °C would act to decrease or increase the estimated changes in probability ratio and magnitude, our conclusions would remain unchanged.

Since the usage of GWI is standard practice in WWA analyses, there are many examples of other studies using this index,, such as the study on the 2021 Pacific Northwest heatwave (Philip et al., 2022),  the 2022 India-Pakistan spring

heatwave (Zachariah et al., 2023), or the ensuing heavy rainfalls and floods in Pakistan in the 2022 summer (Otto et al., 2023).

5.  L355 Fit of Gaussian to soil moisture residuals to covariate. It is stated here that all the "variable of interest are reasonably Gaussian distributed" but I can't find a goodness of fit assessment other than what is implicit in the return level figures. (Note this is a different matter to how well the fitted parameters agree between observations-based products and models.) Looking at the fit uncertainty lines displayed in, for e.g., Fig. 4 I'm not very concerned that the tails are badly represented, though a comment on the fit shown in the left hand panel of Fig. 5(a) (ERA5-Land, NHET root-zone) might be due as there is obvious non-trivial behaviour suggested in the residuals to the covariate.

    First, the reviewer is right in that we do not provide other means for assessing the goodness of fit than the return level figures. Nevertheless, the WWA workflow includes a thorough assessment whether a given distribution is a good fit (e.g., by also considering Q-Q plots; citing Philip et al., 2020):

    *"Depending on the interest field, several distributions can be tested on how well the sample data are fitted by these different distributions by evaluating a quantile–quantile plot or, equivalently, a return time plot that shows the same information but emphasizes the tail. Gaussian distributions are often seen not to describe the tails well. A choice is made based on the comparison."*

    As stated above, the misuse of a Gaussian distribution is often most apparent for the tails, which does not seem to be a problem for our analysis. Therefore, and considering the fact that we analyse spatiotemporally large averages, we believe our choice of statistical distribution is adequate.

    Second, we appreciate that the reviewer asks about the "non-trivial" behaviour shown in Fig. 5a, which likely refers to the fact that observations tend to cluster below [< 0 °C | >0.6 °C] or above [> 0 °C & < 0.6 °C] the best estimate of the location parameter). To explore the likely causes, we present **Figs. R1** and **R2**:

[Figure]

**Fig. R1:** Summer mean soil moisture for the northern extratropics, identical to Fig. 1b from the manuscript and pasted here for a more convenient comparison to Fig. R2.

[Figure]

**Fig. R2:** Summer mean temperature and precipitation averaged over the land masses of the northern extratropics, based on ERA5 (1950–2022). The warming stalled until the 1980s due to aerosol emissions (e.g., Wild et al., 2007), and the precipitation timeseries evidences variability at decadal and multidecadal timescales yet no clear trends across the entire available time period.

When visually comparing the ECMWF soil moisture estimates to these ERA5 temperature and precipitation timeseries above, it may seem that only precipitation matters (which indeed exerts strong control on interannual variability). Throughout the entire timeseries, there is no clear decline in precipitation (which is confirmed by our trend analysis), but there seems to be

an increase until right about the year 2000, followed by a noticeable jump around 2000/2001. According to Hersbach et al. 2020, this decline is not in agreement with observations from GPCC and GPCP (Fig. 23 therein) and might be an artefact. However, the fact that the amount of precipitation is quite comparable in the first 20 and last 20 years of the timeseries also suggests that the overall soil moisture changes are not predominantly precipitation-driven, but rather a consequence of the strong warming. However, due to the lack of warming in about the first half of the timeseries, which is accompanied by precipitation increases, NHET root-zone soil moisture in both ERA5 and ERA5-Land is likely dominated by precipitation rather than GMST (which slightly increases in this period), and actually increases. At some point, and this appears to be once precipitation drops around 2000 and then remains rather stable, the drying signal emerges clearly, fuelled by strong warming.

Since we model changes in intensity (or the location parameter) solely in dependence of GISTEMP GMST and hence do not capture the (multi-)decadal variability of ERA5 precipitation, the residuals in Fig. 4a (left hand side) are mainly found above the location parameter for GMST anomalies (with respect to 1951–1980, which we now also mention in the description of GISTEMP ) between 0 °C to about 0.6 °C.

6. L369 Here it is stated that the Gaussian for precipitation scales with GMST, while in the caption to Figure S3 states constant dispersion parameter (which it would be helpful to define). Meanwhile evaluation tables S1 – S4 state a single value of sigma for precipitation. Given that the large area mean precipitation is being modelled as a Gaussian I am not entirely sure what to expect but a simple shift with GMST, like the temperature, might be reasonable. Indeed Fig. S3 left panels seem to suggest a simple shift with GMST. Which of these three options (just scale, shift and scale or just shift) is being applied to the precipitation in this analysis? Is this a deviation from or addition to the WWA protocol?

We use a Gaussian distribution that, following the WWA terminology (Philip et al., 2020), "scales" with GMST for precipitation and soil moisture in our analysis, which means both the location and scale parameters depend on GMST (but their ratio, the dispersion parameter, is assumed to remain constant). (As a side comment, in Philip et al. (2020), the term "shifting *and* scaling" is used to

distinguish an approach in which the location and scale parameters are estimated independently, unlike for the regular "scaling"). We are grateful that the reviewer spotted a typo in evaluation tables S1 – S4; for precipitation, we indeed provide the dispersion parameter, and not sigma. We also noticed that not all the data was copy-pasted correctly from our analysis sheets to create tables S2–S4, and have corrected all resulting errors. Our analysis is not affected by this.

As for the decision to model precipitation with a distribution whose location *and* scale parameters depend on GMST, each modelled with an exponential function depending on GMST similar to the exponential dependence of saturation vapour pressure on air temperature (Philip et al., 2020), this assumption has been employed and validated in earlier studies (e.g., Otto et al., 2018; Philip et al., 2018.). Despite the use of an exponential function, the resulting fit of the location (and scale) parameters to GMST is nearly linear. This explains why In early stages of the study, we repeated part of the attribution analysis for soil moisture with a Gaussian shifting approach (as employed for temperatures), and obtained similar results. Additionally, following the WWA protocol, we evaluate for all observational datasets and models whether the assumptions required for the statistical model hold; to provide an example, for precipitation, we check the dispersion parameter calculated for (15-yr) moving windows (**Fig. R3**) .

[Figure]

**Fig. R3:** Running dispersion of summer (JJA) mean precipitation for WCE, based on ERA5. There is no clear long-term trend in the dispersion parameter, and hence we deem our assumption of a constant dispersion ratio appropriate. Produced with the Climate Explorer.

7.  L386 The phrase "natural variability" refers to internally generated variability here. I appreciate that this phrase is in conventional use (and used as such in the

protocol paper, Philip at al., 2020) but always feel it is bound to cause confusion with "natural variability" from solar and volcanic external forcing, which is most certainly not internally generated. Maybe a parenthesis "(internally generated)" or some such would be helpful to readers familiar with other uses of the term "natural".

We believe the reviewer's suggestion could indeed prevent confusion among readers, and include "(internally generated)". The corresponding section has been edited based on reviewer #2's question 6, and is pasted in our response further below in this rebuttal.

8. L604 Only reference to important begged question: why is the choice of covariate GMST rather than NHET or WCE mean JJA mean temperatures? Is there a physical motivation the authors can make that global annual mean temperature is an appropriate covariate to choose? Without some defence of this choice we could imagine that a soil moisture response to a different temporal evolution in JJA could change the results. I have taken a look at NHET mean JJA mean temperatures (on Climate Explorer, using GISS 1200km monthly fields) and can see that the temporal evolution is so similar to annual mean GMST that I doubt using this as a covariate would make any difference to the results, so I am not concerned, but a brief defence should be made.

The main reason for using GMST as covariate is that its increase since pre-industrial times has been attributed to human influences, since natural factors only account for ± 0.1 °C since 1850–1900 (Eyring et al., 2021), and thus serves as a reliable indicator for anthropogenic climate change (Gudmundsson & Seneviratne, 2016; Wartenburger et al., 2017; Quilcaille et al., 2021).

We also note that the mean temperature increases of a 'small' region such as WCE may not only indicate the effects of anthropogenically induced climate change (i.e., the response to an external forcing), but can also be impacted by atmospheric circulation changes, which in turn are considered to be manifestations of (multi-decadal) internal variability. At very large (hemispheric and global) scales, the warming signal emerges more clearly, and we apply a 4-year running mean to GMSTs to dampen the influence of El Niño (Philip et al., 2020). The practice of regressing a local (or regional) climate variable onto a global climate variable, typically GMST, is not only extremely common in

attribution science (all attribution studies cited in this rebuttal use GMST as a covariate), but also forms the backbone of pattern scaling (e.g., Tebaldi & Arblaster, 2014).

Therefore, by statistically exploiting the relationship between summer mean soil moisture and GMST, we arrive at estimates of how human influences shape the probability and magnitude of soil drought. Our approach allows us to communicate our results effectively: we can make attribution statements that directly relate to the pre-industrial, comparatively less human-influenced climate, as well as future climate states (here at a global warming level of 2.0 °C).

9. L629 This section is very brief and you have to strain your eyes a little to see the origin of the numbers "2-30 fold" for projections in Fig. S13 – S16. I can see combined model best estimates from around 1.5 to what may be around 30 in Fig. S16 if I read the scale right. A statement of the values in the text would make this more transparent. It might also be worth noting that, again, PR lower bounds include 1 in every case.

We agree that especially for Figs. S15–S16 (owing to the wide uncertainty bars and hence plotting ranges), the visual extraction of values such as the best estimates is challenging, and thank the reviewer for bringing this to our attention. We now provide the best estimates of the probability ratios shown in Fig. S13 – S16 in the main text:

For all event definitions, a further increase in intensity as well as a ~2–30-fold further increase in the frequency of such an event is found (**Suppl. Figs. 13–16**): for WCE (NHET) and in terms of best estimates, the PR of root-zone and surface soil moisture droughts amount to 1.6 and 2.3 (14.6 and 32.4), respectively.

10. L629 §11.6.1.3 of AR6 (AR6 chapter 11) points out that "There is evidence [in ESMs] that surface soil moisture projections are substantially drier than total soil moisture projections, and may overestimate drying of relevance for most vegetation", citing Berg et al., (2017). This contrasts with the state of affairs pointed to in L610 pertaining to historical times described as "largely consistent". I note that the largest PR in the projections is for surface, NEHT, perhaps suggesting that the models used here concur with this expectation. Is this the case?

We thank the reviewer for pointing out this apparent inconsistency. Berg et al. (2017) investigate soil moisture projections of CMIP5 models, and interpret the gradient between surface and deeper soil moisture , with stronger drying near the surface, as the consequence of "the physical asymmetry between winter precipitation/inflitration" and summer evaporation. If this is the case, we would indeed expect to see stronger surface drying in our observation-derived data, although we also remark that Berg et al. (2017) compare 2070-2099 to 1976–2005, whereas we rely on 1950-2022 to state that the temporal evolution of soil moisture has been "largely consistent"  between the surface and root-zone. Using only ERA5-Land and GLDAS-CLSM, the main observation-based datasets used for the attribution of root-zone and surface soil moisture drought, we express surface and root-zone soil moisture as changes with respect to the mean state of 1950–1980:

[Figure]

**Fig. R4:** ERA5-Land and GLDAS-CLSM surface and root zone soil moisture, expressed in changes (%) with respect to the 1950–1980 average. Using the entire available time series to center the data yields similar results; whereas GLDAS-CLSM points to slightly stronger surface drying in NHET compared to the root-zone, only minor differences emerge for ERA5-Land as well as both datasets and WCE.

A visual comparison of the root-zone and surface soil moisture changes (**Fig. R4**) suggests that the decline in soil moisture is either very similar, or slightly stronger for the surface in the case of NHET and GLDAS-CLSM. Nevertheless, we do not consider this sufficiently strong evidence to confirm the stronger surface

soil drying noted by Berg et al. (2017) for climate projections. We also note that CMIP6 models (and hence most likely also earlier generations such as CMIP5) tend to rely too much on shallow compared to root-zone soil moisture to extract water for transpiration (e.g., Dong et al., 2022; Zhao et al., 2022), and as such, the vertical gradient found by Berg et al. (2017) could be aggravated by model deficiencies.

We added a brief discussion to the manuscript, and plan to include a figure very similar to Fig. R4 in the Supplementary Information:

Nevertheless, the overall temporal evolution of summer soil moisture in the surface layer and root-zone is consistent in both regions for all main datasets, which is plausible given that soil moisture near the surface and in deeper layers is inherently connected through infiltration and diffusion processes (e.g., Albergel et al., 2008). Considering that Berg et al. (2017) reported stronger surface drying than in deeper soil layer for CMIP5 projections, we also compare the historic long-term changes of surface and root-zone soil moisture by representing the respective timeseries as percentage changes for both domains and GLDAS-CLSM and ERA5-Land (**SFig. X**). While there is a stronger decrease in surface than root-zone soil moisture for NHET based on GLDAS-CLSM, comparatively minor drying gradients between the surface and root-zone emerge for NHET using ERA5-Land, and similarly for WCE with both datasets. Our findings thus suggest that soil moisture decreased more near the surface than in deeper layers during the 1950-2022 period, yet the extent of this surface drying gradient remains unclear and might be negligibly small.

11. L634 Despite stressing that the study uses three different types of models (coupled, atmosphere-only and high resolution) there is no discussion of what these different model types have contributed and whether the evaluation and results appear to vary between model types. I appreciate the analysis is already complex but a comment on whether there is anything obvious going on would help. Referring back to my comment on L313, as these different models are all being treated under the same framing then perhaps this lessens the importance of combining different model types. It is also relevant to my comment to L629 concerning the separation of surface and soil moisture projections in ESMs.

We refrain from a detailed discussion of how the different model types fare with respect to their evaluation and results for a multitude of reasons; first, the HighResMIP SST-forced model ensemble unfortunately does not provide root-zone soil moisture, which is the main focus of our analysis, and is therefore only used for the surface soil moisture analysis.

Second, for WCE and root-zone soil moisture, the two AM2.5C360 and FLOR models (with 10 ensemble members each) from the second multi-model ensemble and most of the 20+ CMIP6 models did not pass the validation, and the outcome for surface soil moisture and the same domain is similar. Because we employ a strict model validation based at least on temperature and precipitation, and also soil moisture when available (in spatially explicit form and year-round, as is the case for the CMIP6 models), it is not surprising that for WCE, only a few CMIP6 models yet none of the (far fewer) models part of the 2 other ensembles pass the validation. A systematic analysis would be required to find out if the CMIP6 multi-model ensemble actually performed 'better' in terms of validation.

Third, we believe such an in-depth analysis is beyond the scope of our study, which we agree is already complex. Consequently, since the results are exclusively and largely based on observation-derived estimates and validated CMIP6 models for WCE and NHET, respectively, it is also difficult to comment on how the results might differ across the different model types.

Since all the different models are indeed treated under the same framing, which is now hopefully conveyed more clearly in the main text (see our response to question 1), we believe our choice of considering simulations from different model types for validation & potentially analysis, each of which should in principle be able to capture the relevant physical processes and interactions that lead to summer soil drought, makes the study more robust.

12. L640 PR of "summer drought" in WCE is "about 5". Is this a rounding down of the mean over synthesised best estimates of the 1850- and 1950- WCE root-zone analyses (0.5*(2.8 + 8.8) = 5.8)?

Yes, that is indeed how we arrived at the PR of "about 5". Note that even though we have updated the GLDAS-CLSM dataset (which only affects part of the summer 2022 data rather than the entire time series; see also our response to the other reviewer's first question), the associated probability ratios remain so high that we limit them to 10'000 for the synthesis, and therefore our synthesised best estimates of the PR are still 2.8 (1850–) and 8.8 (1950–). We have adjusted our text to make it more clear where the "about 5" comes from:

Our analysis suggests that the large uncertainties, also due to a lack of in-situ root-zone soil moisture observations except for a few hundred stations, make it difficult to communicate precise numbers. Nevertheless, but the synthesised probability ratio for a 2022-like summer drought in West-Central

Europe is likely larger than 1 and amounts to about 5 (2.8 when using 1850–2022 model data, and 8.8 for 1950–2022).

13. L643 PR inc. 1, "this also applies to the northern extratropics" – but Fig. 8 shows that NHET root-zone PR has both lower bounds > 1.

Yes, the sentence is indeed misleading, thanks for spotting this. We have revised the text:

This also applies toFor the northern extratropics, where our analysis suggests a stronger overall warming signal, — with a probability ratio of at least 20, — and associated decline in root-zone soil moisture.

14. L668 Again the origin of the numbers requires a bit of work, given that brief section 5.5 simply stated a single PR range of 2 – 30 concerning four analyses. I think that the numbers '10 years' and 'every year' have been arrived at by dividing 20 year return times for each region by the average of PR estimates over surface and root-zone best estimates (about 2, and about 20). Is this the case?

This is the case, yes, we converted our results to changes in the return period using rough numbers. The accurate averages of surface and root-zone soil moisture drought PRs for WCE and NHET are (1.6 + 2.3) / 2 = 1.95 and (14.6 + 32.4) / 2 = 23.5, respectively. We believe communicating rough values is appropriate here, since a PR of, e.g., "1.95" could convey a higher impression of accuracy than is justified for our analysis. We added an additional sentence such that it hopefully becomes more clear where the "every 10 years" and "every single year" come from:

Moreover, the models analysed also show that soil moisture drought will continue to increase with additional global warming — in West-Central Europe, a 2022-like event or worse is expected to occur about every 10 years once a warming level of 2 °C is reached, and nearly every single year in the northern extratropics. In other words, for 0.8 °C additional warming compared to the present, the mean probability ratios of surface and root-zone soil moisture drought in West-Central Europe and the northern extratropics amount to about 2 and 20, respectively.

Technical corrections

L115 "crushed" – typo?

Thanks, we have rephrased the sentence as follows:

[...] and by late June, Italy had surpassed its historical wildfire average threefold.

L240 "of" à "over" or "for" would be better?

We appreciate this suggestion and have edited the sentence:

The E-OBS dataset is a 0.25° × 0.25° gridded temperature and precipitation dataset for Europe, [...]

L337 Statistical methods: this section could do with a sentence stating that the 2022 event is characterised by taking the return time of the event in the observations-based products and then querying the model distributions at the corresponding return level. Those already familiar with the WWA method will know already but I can't find that this essential aspect of the method is explicitly stated anywhere. It may also then make more sense why later in L476 we are happy to take the two estimates for the observations based return times and simply average and round them.

Thank you for pointing this out, we agree that such a sentence is helpful for readers unfamiliar with the WWA approach and have, based on the reviewer's suggestion, edited the section:

The essence of the approach we employ here is that event indices — regional summertime averages of soil moisture, precipitation, temperature — are represented with continuous probability distributions conditional on GMST, which enables us to estimate how the event magnitude and probability of occurrence have changed under human-induced climate change. We characterise the 2022 summer drought by first determining the return time of the event with the observations-based products and then querying the model distributions at the corresponding return level. The analysis steps include: [...]

L401 Typo: I think "Note that as to" should be "Note that so as to".

Thank you, the missing "so" has been inserted.

L483 Type: I think "inform on" should probably be "concern"

We believe "inform on" is not wrong since the intensity changes we communicate, using a simple example, essentially "contain" the same information as the slope of the linear regression of the index onto GMST: e.g., the roughly 1.8 °C of synthesised intensity change for WCE summer temperatures (present-vs-past) imply a slope of ~1.5 (1.8 °C regional warming / 1.2 °C GMST change). Nevertheless, to avoid confusion since we do

indeed first determine the slope (and other parameters) and only then obtain the mean intensity changes, we rephrase this sentence:

[...] of mean intensity changes. The latter are less sensitive than the probability ratios to the inferred relationships between global warming and long-term soil moisture changes, since they  are derived from the linear trend between the covariate (here GMST) and the index (here regional summer mean root-zone soil moisture) rather than the ratio of occurrence probabilities, for which the denominator — the probability of the event for the past climate — can become very small, as is the case for GLDAS-CLSM.

L497 I had to look "Mio" up, it being short hand for 1 million in German. Might best be just "million"?

Thank you for the suggestion, we now use "million" instead.

L499 Type: "2002" should be "2022".

Thanks for spotting this typo.

L504 "excessive" seems a strange adjective. Suggest "very large"?

We also gladly accept this suggestion, thank you.

L548 repeated use of "fairly" is ambiguous language.

We agree that some editing is required here, and change the text as follows:

Consequent,  the model probability ratios are more consistent with ERA5-Land, although GLDAS-CLSM still features a much stronger warming signal. In terms of the more robust changes in intensity, GLDAS-CLSM is  similar to MPI-ESM1-2-LR, [...]

L583 "temperature surpluses" is a strange phase, something like "positive temperature anomalies" would be better.

We agree that 'surplus' makes more sense for a variable like precipitation. We use "positive temperature anomalies" now.

L592 Apparent error in statement of uncertainty bounds for NHET change in intensity compared to Fig. S9.

Thanks for catching this error, the text is now consistent with Fig. S9 (which has been correct in the first place):

Similarly, for the northern extratropics, the change in intensity is 1.9 (1.7 to 2.1) °C.

L614 "Appendix" here I assume refers to the SI, Fig. S17 and S18.

Thank you, this relic has been removed and the text now refers to Fig. S17 and S18.

L629 Typo: "0.8oC" should be "0.8°C"

Thanks, fixed.

L649 declining root-zone soil moisture in the 21st century in "the entire northern extratropics" – we don't have a figure showing this. Perhaps a reference to the literature?

We think this sentence may have been misleading, as we intend to communicate that *the average* soil moisture in the northern extratropics has declined. This does, however, not imply soil drying *everywhere* in the northern extratropics, hence the use of "entire northern extratropics" is not ideal. We thus changed the sentence:

Similarly, nearly all CMIP6 models display declining summertime root-zone soil moisture throughout the 21st century in West-Central Europe and averaged over the northern extratropics, [...]

Table S6 units for temperature intensity change: are the numbers really in % or are they °C? The numbers seem consonant with L489.

Another great catch, thank you for noticing this typo. The numbers are indeed in °C.

The quality of some figures is too low, e.g. Fig. S9 and S10, perhaps Figures 4 and 5 as the 2022 data point is hard to see.

We agree, and will improve the quality of the mentioned figures in the revised manuscript and supplemetary information.

Table S4 is for surface soil moisture but the column heading says root-zone.

Thanks for noticing this error and being such an attentive reviewer.

References

AR6 chapter 11: Seneviratne, S.I., X. Zhang, M. Adnan, W. Badi, C. Dereczynski, A. Di Luca, S. Ghosh, I. Iskandar, J. Kossin, S. Lewis, F.  Otto, I.  Pinto, M. Satoh, S.M. Vicente-Serrano, M. Wehner, and B.

Zhou, 2021: Weather and Climate Extreme Events in a Changing Climate. In Climate Change 2021: The Physical Science Basis. Contribution of Working Group I to the Sixth Assessment Report of the Intergovernmental Panel on Climate Change [Masson-Delmotte, V., P. Zhai, A. Pirani, S.L. Connors, C. Péan, S. Berger, N. Caud, Y. Chen, L. Goldfarb, M.I. Gomis, M. Huang, K. Leitzell, E. Lonnoy, J.B.R. Matthews, T.K. Maycock, T. Waterfield, O. Yelekçi, R. Yu, and B. Zhou (eds.)]. Cambridge University Press, Cambridge, United Kingdom and New York, NY, USA, pp. 1513–1766, doi:10.1017/9781009157896.013

Berg, A., J. Sheffield, and P.C.D. Milly, 2017: Divergent surface and total soil moisture projections under global warming. Geophysical Research Letters, 44(1), 236–244, doi:10.1002/2016gl071921.

Climate Explorer: https://climexp.knmi.nl/start.cgi

Philip, S., Kew, S., van Oldenborgh, G. J., Otto, F., et al.: A protocol for probabilistic extreme event attribution analyses, Adv. Stat. Clim. Meteorol. Oceanogr., 6, 177–203, doi:10.5194/ascmo-6-177-2020, 2020.

**References**

Berg, A., J. Sheffield, and P.C.D. Milly, 2017: Divergent surface and total soil moisture projections under global warming. *Geophys. Res. Lett.* **44**, 236–244. doi:10.1002/2016gl071921

Dong, J., Lei, F. and Crow, W.T. (2022): Land transpiration-evaporation partitioning errors responsible for modeled summertime warm bias in the central United States. *Nat. Commun.* **13**, 336. doi:10.1038/s41467-021-27938-6

Eyring, V., N.P. Gillett, K.M. Achuta Rao, R. Barimalala, M. Barreiro Parrillo, N. Bellouin, C. Cassou, P.J. Durack, Y. Kosaka, S. McGregor, S. Min, O. Morgenstern, and Y. Sun, 2021: Human Influence on the Climate System. In *Climate Change 2021: The Physical Science Basis. Contribution of Working Group I to the Sixth Assessment Report of the Intergovernmental Panel on Climate Change* [Masson-Delmotte, V., P. Zhai, A. Pirani, S.L. Connors, C. Péan, S. Berger, N. Caud, Y. Chen, L. Goldfarb, M.I. Gomis, M. Huang, K. Leitzell, E. Lonnoy, J.B.R. Matthews, T.K. Maycock, T. Waterfield, O. Yelekçi, R. Yu, and B. Zhou (eds.)]. Cambridge University Press, Cambridge, United Kingdom and New York, NY, USA, pp. 423–552, doi:10.1017/9781009157896.005

Gudmundsson, L. and Seneviratne, S. I. (2016): Anthropogenic climate change affects meteorological drought risk in Europe. *Environ. Res. Lett.* **11,** 044005. doi:10.1088/1748-9326/11/4/044005

Otto, F. van der Wiel, K., van Oldenborgh, G. J., Philip, S.,Kew, S. F. *et al*. (2018): Climate change increases the probability of heavy rains in Northern England/Southern Scotland like those of storm Desmond—a real-time event attribution revisited. *Env. Res. Lett.* **13**, 024006. doi:10.1088/1748-9326/aa9663

Otto, F., Zachariah, M., Saeed, F.Siddiqi, A. *et al*. (2023): Climate change increased extreme monsoon rainfall, flooding highly vulnerable communities in Pakistan. *Environ. Res.: Climate* **2,** 025001. doi:10.1088/2752-5295/acbfd5

Philip S., Kew, S. F., van Oldenborgh, G. J., Aalbers, E., Vautard R. *et al.* (2018): Validation of a Rapid Attribution of the May/June 2016 Flood-Inducing Precipitation in France to Climate Change. *J. Hydrometeorol.* **19**, 1881–1898. doi:10.1175/JHM-D-18-0074.1

Philip, S., Kew, S., van Oldenborgh, G. J., Otto, F., Vautard, R., van der Wiel, K., King, A., Lott, F., Arrighi, J., Singh, R., and van Aalst, M. (2020): A protocol for probabilistic extreme event attribution analyses, *Adv. Stat. Clim. Meteorol. Oceanogr.* **6**, 177–203. doi:10.5194/ascmo-6-177-2020

Philip, S. Y., Kew, S. F., van Oldenborgh, G. J., Anslow, F. S. *et al. (2022)*: Rapid attribution analysis of the extraordinary heat wave on the Pacific coast of the US and Canada in June 2021. *Earth Syst. Dyn.* **13**, 1689–1713 doi:10.5194/esd-13-1689-2022

Quilcaille, Y., Gudmundsson, L., and Seneviratne, S. I. (2023): Extending MESMER-X: A spatially resolved Earth system model emulator for fire weather and soil moisture. *EGUsphere* [preprint], doi:10.5194/egusphere-2023-589

Tebaldi, C., Arblaster, J.M. (2014): Pattern scaling: Its strengths and limitations, and an update on the latest model simulations. *Climatic Change* **122**, 459–471. doi:10.1007/s10584-013-1032-9

Wild, M., Ohmura, A., Makowski, K. (2007): Impact of global dimming and brightening on global warming. Geophys. Res. Lett. 34, L04702. doi:10.1029/2006GL028031

Zachariah, M., Arulalan, T., AchutaRao, K., Saeed, F., Jha, R. *et al* (2023): Attribution of 2022 early-spring heatwave in India and Pakistan to climate change: lessons in assessing vulnerability and preparedness in reducing impacts. Environ. Res.: Climate, in press. doi:10.1088/2752-5295/acf4b6

Zhao, M., A, G., Liu, Y. *et al.* Evapotranspiration frequently increases during droughts. *Nat. Clim. Chang.* 12, 1024–1030 (2022). doi:10.1038/s41558-022-01505-3

Wartenburger, R., Hirschi, M., Donat, M. G., Greve, P., Pitman, A. J., and Seneviratne, S. I. (2017): Changes in regional climate extremes as a function of global mean temperature: an interactive plotting framework, Geosci. Model Dev., 10, 3609–3634, doi:10.5194/gmd-10-3609-2017

---

## Author Comment (AC2)

**REVIEW #2**

The author conducted attribution analysis of a soil moisture drought event in West-Central Europe in 2022 based on a large number of soil moisture estimates. It is found that the return period of the drought event is decreased substantially in current climate as compared with that in pre-industrial climate, although there are uncertainties in soil moisture estimates. The manuscript is well written and easy to follow, and there are only a few minor comments below.

We appreciate the positive assessment of our work and the questions listed below, to which we reply individually in blue font.

1.  GLDAS_ CLSM data includes v2.0 (1948-2014) and v2.1 (2000-2022). Is v2.0 and v2.1 used together in the article? How are they merged? Have corrections been made before merging? Only the June data was used in 2022? How about July and August?

    Thank you for this comment; in short, we have repeated much of our analysis for an updated GLDAS-CLSM dataset, and explain the merging procedure in the following. We use GLDAS-CLSM data forced with Princeton v2.0 meteorological forcing and without data assimilation (i.e., open loop, available for 1948–2012), and then ECMWF-analysis forced CLSM output obtained with GRACE data assimilation from 2003 onwards. The latter is scaled to Princeton-forced data based on 2003–2012 (that is, mean and standard deviation are matched), employing a 7-day moving window. Since GRACE data was not available for July and August 2022 when we performed the initial analysis, CLSM was originally run in open-loop mode for those months instead (but still driven by the ECMWF analysis fields).

    We have since updated our CLSM dataset, such that the entire summer of 2022 is now based on CLSM with GRACE data assimilation. For both WCE and NHET, the updated mean June–August 2022 soil moisture is slightly higher than the initial version without GRACE data assimilation for July and August 2022, but the differences are so small that they are not visible in most figures. A noteworthy exception concerns the event return periods, which remain nearly unchanged for NHET (root-zone: 7 years→ 6 years, surface: 17 years → 14 years), but

decrease a bit for WCE (root-zone: 33 years → 23 years, surface: 47 years → 33 years), consistent with the slightly higher summer 2022 soil moisture (i.e., a bit less dry and hence less extreme). Since this only results in minor changes of the average 2022 soil moisture drought return periods (always based on GLDAS-CLSM and ERA5-Land, and additionally ERA5 for surface soil moisture), we keep a return period of 20 years for the models in all our soil moisture analyses.

Even though the probability ratios of GLDAS-CLSM decreased, they are still (more than) high enough to be limited to 10'000 (as described in the Methods). Consequently, the PRs in the present-vs-past synthesis figures remain unchanged. The more robust intensity changes of GLDAS-CLSM differ only slightly from our initial analysis, such that the synthesised values barely change compared to our first analysis.

We updated all the text, figures and tables that relate to or make use of GLDAS-CLSM data, and edited the description of GLDAS-CLSM to hopefully make the scaling procedure more clear:

The 2003 to present data is scaled  to the open loop, using scaling factors determined for each grid cell and with a 7-day moving window such that the mean and standard deviation of soil moisture obtained with GRACE data assimilation across 2003–2012 matches the Princeton-forced (1948-2012) climatology in the same period  (Houborg et al., 2012). For some grid cells in high latitudes, this results in negative and hence not physically meaningful values, which we remove for our analysis of the northern extratropics. ~~Note that the GLDAS-CLSM dataset employed here was produced with GRACE/GRACE-FO data up to and including June 2022. We do not expect this to affect our conclusions, since (i) the attribution statements ultimately depend on the linear trend in soil moisture as a function of global warming for the entire period of 1950-2022, and (ii) the return periods of the 2022 event are similar in ERA5-Land and GLDAS-CLSM.~~

2. Why does CMIP6 use SSP585 and historical scene splicing (usually SSP245), while CMIP5 uses RCP4.5 and historical splicing?

We appreciate the reviewer's question and remark that the choice of future scenario is not as important as it may seem in our analysis framework, although this was not clearly described in the initial manuscript. We employ global warming levels, -1.2 and +0.8 °C with respect to 2022 in our analysis. Therefore, it is not problematic if we fit our statistical model for a GMST range slightly larger than what we ultimately need for our evaluation, which is in fact even desirable

(unlike the opposite case, i.e. fitting the statistical model for say -1.2 to only +0.1 °C, and then querying the fitted model at +0.8 °C).

We have edited Sect. 3.3 as follows to make our approach more clear:

We note that regardless of the underlying emission scenario, model data from 1850 to 2022 and from 1850 to 2050 are used to conduct the present-vs-past and future-vs-present climate analyses, but these time periods only indicate the amount of data used to fit the statistical model and hence infer the relationship between event indices and GMSTs. We then rely on global warming levels to calculate the return periods, the probability ratio (PR — the factor-change in the event's probability) and change in intensity of the drought event. . For our comparison of the present (2022) to the past (1850–1900) climate, the GMST changes with respect to the present amount to -1.2 ° C according to the Global Warming Index https://www.globalwarmingindex.org), and for comparing additional changes in the future to the present, we use +0.8 ° C relative to the 2022 GMSTs (+2.0 ° C with respect to pre-industrial conditions). As such, it does not matter when the future warming is reached in any given model simulation, which allows us to combine models with different emission scenarios and still perform a consistent analysis.

3. The uncertainty range for the surface and rootzone soil moisture anomalies (Figures 1a and 3a) could be shown in the supplement materials.

   Only GLDAS-CLSM is a good candidate to supplement Fig. 1a (global 1950–2022 data available), and the anomaly patterns are generally similar to ERA5-Land (used for Fig. 1a). We provide a comparison below for root-zone soil moisture (cf. **Fig. R5 & R6**):

[Figure]

**Fig. R5:** 2022 summer root-zone soil moisture anomalies based on GLDAS-CLSM and using 1950–2022 as a baseline.

[Figure]

**Fig. R6:** Fig. 1a from the manuscript; like Fig. R5, but using ERA5-Land.

Besides West-Central Europe, where both datasets point to widespread negative anomalies, drought conditions were also reported by other regions, such as the central and southwestern United States or China. Both ERA5-Land and GLDAS-CLSM indicate soil moisture deficits in these areas.

We also show GLDAS-CLSM based surface soil moisture anomalies below (**Fig. R7**), which also does not agree with ERA5-Land everywhere, but certainly supports the main message of Figs. 1a and 3a: West-Central Europe experienced a strong soil drought in the summer of 2022, and despite positive soil moisture anomalies in regions such as northeastern North America or parts of Siberia, soil moisture deficits also dominated the northern extratropics as a whole.

[Figure]

**Fig. R7:** Fig. 1a from the manuscript; like Fig. R6, but for surface soil moisture.

We believe that the different timeseries in Figs. 1b–c and 3b–c are more efficient to convey the "observational" uncertainties, since they make use of several (and not just two) datasets, and additionally show that the uncertainty decreases in time.

4. What is the definition of Intensity in this study?

The intensity corresponds to the mean event magnitude, which changes as our climate warms. In other words, climate change affects the expected magnitude of an event with any given return period (which is a different but equally viable perspective as considering how the probability of an event with a given magnitude changes). Technically, the intensity is represented by the location parameter of the scaling and shifting Gaussian distributions that we employ, which in turn is modelled as a function of global mean temperature changes, and has the same units as the index of interest (e.g., summer mean root-zone soil moisture).

We use magnitude and intensity synonymously throughout the text, and have edited the first occurrence as follows to hopefully make it more clear:

The essence of the approach we employ here is that event indices — regional summertime averages of soil moisture, precipitation, temperature — are represented with continuous probability distributions conditional on GMST, which enables us to estimate how the intensity (event magnitude) and probability of occurrence have changed under human-induced climate change.

5.  Line 479: Does -9% (-13%.. -4%) refer to a change in intensity? Please explain in detail.

    Yes, this does refer to a change in intensity. As stated in our response above, the intensity or event magnitude has the same units as the respective index. We communicate changes in intensity in °C for temperature, and use percentages for precipitation and soil moisture. This is motivated by the fact that the implications of absolute temperature changes are intuitive, whereas relative changes can easily be interpreted even without knowledge of the respective total (or, in this case, past climate state) such as, e.g., the mean volumetric root-zone soil moisture of the northern extratropics.

    We edited the text to make it more clear that we do indeed refer to a change in intensity:

    We also estimate the mean change in WCE summer root-zone soil moisture from the past to the present climate, and obtain which yields intensity changes with best estimates (confidence intervals) of -9% (-13% .. -4%) for ERA5-Land and -14% (-16% .. -11%) for GLDAS-CLSM.

6.  What is the physical meaning of representation error? Perhaps only when the minimum value of Representation error is significantly greater than 1 (less than 0) can the impact of anthropogenic climate change on Probability ratio (change in intensity) be considered significant?

In the WWA framework, observations — or in the case of this study, observation-derived estimates (of soil moisture) — are considered as equally valid representations of the same climate realisation (i.e., reality), and hence sample the same (true) underlying natural variability. If the observations were perfect, we would expect them to indicate identical best estimates, but this is clearly not the case. Therefore, the mean discrepancy of individual best estimates of observations to the observational mean best estimate serves as a "representation" error.

In the context of the last question (8.), we would like to emphasise here that the observation-derived soil moisture estimates are associated with a large representation error compared to temperature or precipitation (cf. Fig 6 or 7 and Fig. S7, S8). This error is primarily large not because the datasets do not agree on the presence of a clear warming footprint in soil moisture, but rather because they either point to a signal consistent with the models (ERA5-Land) or an even stronger one (GLDAS-CLSM).

We thank the reviewer for their question and think that our revised text pasted below makes the approach more clear:

To combine the two lines of evidence into a synthesised assessment, first, a representation error is added (in quadrature) to the observations, to account for the difference between observations-based datasets that cannot be explained by natural variability (light blue bars). This is The rationale behind this is that we consider observations as equally valid representations of a singular climate realisation with the same underlying true natural (internally generated) variability. Therefore, the mean deviation of individual datasets to the overall mean best estimate indicates a representation error (of observations with respect to reality), shown in the synthesis figures as white boxes around the natural — that is, internally generated — variability (light blue bars). The dark blue bar shows the average over the observation-based products (black marker) and the total uncertainty (width of the bar) based on natural variability and representation errors. Instead of representation errors, next, a term to account for intermodel spread is added (in quadrature) to the natural variability of the models. Note that while this term is based on the scatter of model means (analogous to the representation error for observations), we interpret model simulations as independent climate realisations. Consequently, we only add this term granted that the differences between models cannot solely be explained by natural variability, which is the case here. ThisThe intermodel spread is shown in the synthesis figures as white boxes around the light red bars.

7. There are a total of 25 models, but only 7 models in Figures 6-7 have passed the test. How about using all models? This may increase the reliability of the results.

Thank you for the suggestion. This study closely follows the WWA approach (Philip et al., 2020) that has been used for numerous rapid attribution and also many peer-reviewed publications. In this detailed protocol, the model validation and subsequent restriction to validated models is pivotal to ensure the main goal of extreme event attribution to human-induced climate change using both observations and climate models: to make the analysis more robust, as climate models have the advantage of providing alternative realisations of climate variability that can supplement observational records (e.g., Perkins & Fischer, 2013). However, since models are necessarily stark simplifications of our complex climate system and their performance depends both on the variable and region of interest, considering models without validation could decrease rather than increase the reliability of an attribution study.

We thus think that our focus should remain on synthesised estimates of probability ratios and changes in intensity. Nevertheless, we already provide a statement in the main text that is based on all (CMIP6) models:

"Finally, we remark that among the 25 available CMIP6 models used here (of which 7 passed the validation), all agree that based on 1950–2022, the best estimate of the probability ratio is at least 1, and oftentimes on the order of 10 or higher."

8. In Figures 6 and 7, the attribution results of the probability ratio are not significantly greater than 1 (the minimum value of the confidence interval is less than 1), which does not seem to suggest that the impact of anthropogenic climate change on drought in 2022 is significant. Please explain.

The probability ratios shown in Figs. 6 and 7 suggest that human influence has most likely increased the probability of the event, but the reviewer is right in that we cannot entirely exclude the possibility that this influence is relatively weak or even renders the event slightly less probable. However, we think that other factors are also relevant here. Firstly, the "observational" uncertainty, which contributes to the synthesised uncertainty together with the model uncertainty (see also response to question 6), is primarily high because the different datasets indicate either a moderate (ERA5-Land, ERA5) or strong (GLDAS-CLSM) warming imprint, and not due to, e.g., opposing signals.

Moreover, the lower bound of the more robust intensity changes is clearly > 0 for Fig. 7 (1950 onwards for models, as for observations), and only very slightly < 0 for Fig. 6.

Furthermore, widespread (summer) soil drying is also clearly expected due to our changing climate: mainly because strong warming increases evaporation which, unless counteracted by changes in precipitation [and/or runoff], causes soil desiccation. Particularly the dependence of soil moisture on precipitation, characterised by large internal variability, greatly increases the uncertainty, whereas the temperature signal is comparatively obvious.

We believe that we already clearly communicate the high level of uncertainty compared to more standard attribution analyses of, e.g., heatwaves. To quote the other reviewer:

> *"The authors are at pains to emphasise the sources of uncertainty in the analysis, not least of which arise from the observations or observations-based products. This uncertainty is explored at length and conclusions have been well phrased in light of this. The authors provide a convincing argument that robust yet conservative conclusions about the change in soil drought can be made and highlight that with lower confidence stronger statements are possible."*

For all the reasons outlined above, we think that our conclusions are not nullified by the fact that the results for WCE point to a low, but non-zero probability that the anthropogenic impact did/does not exacerbate the event's probability of occurrence, and are hence adequate.

**References**

Perkins, S. E. and Fischer, E. M. (2013): The usefulness of different realizations for the model evaluation of regional trends in heat waves. Geophys. Res. Lett. 40, 5793-5797. doi:10.1002/2013GL057833

Philip, S., Kew, S., van Oldenborgh, G. J., Otto, F., Vautard, R., van der Wiel, K., King, A., Lott, F., Arrighi, J., Singh, R., and van Aalst, M. (2020): A protocol for probabilistic extreme event attribution analyses, *Adv. Stat. Clim. Meteorol. Oceanogr.* 6, 177–203. doi:10.5194/ascmo-6-177-2020